# Amino acid catabolite markers for early prognostication of pneumonia in patients with COVID-19

Rae Maeda[1,13], Natsumi Seki[1,13], Yoshifumi Uwamino[2,3], Masatoshi Wakui[2], Yu Nakagama[4], Yasutoshi Kido[4], Miwa Sasai [5,6], Shu Taira[7], Naoya Toriu[8,9], Masahiro Yamamoto [5,6], Yoshiharu Matsuura [5,6], Jun Uchiyama[10], Genki Yamaguchi[10], Makoto Hirakawa[10], Yun-Gi Kim[10], Masayo Mishima[11], Motoko Yanagita [8,9], Makoto Suematsu [11,12] & Yuki Sugiura [1,11] ✉

Effective early-stage markers for predicting which patients are at risk of developing SARS-CoV-2 infection have not been fully investigated. Here, we performed comprehensive serum metabolome analysis of a total of 83 patients from two cohorts to determine that the acceleration of amino acid catabolism within 5 days from disease onset correlated with future disease severity. Increased levels of de-aminated amino acid catabolites involved in the de novo nucleotide synthesis pathway were identified as early prognostic markers that correlated with the initial viral load. We further employed mice models of SARS-CoV2-MA10 and influenza infection to demonstrate that such de-amination of amino acids and de novo synthesis of nucleotides were associated with the abnormal proliferation of airway and vascular tissue cells in the lungs during the early stages of infection. Consequently, it can be concluded that lung parenchymal tissue remodeling in the early stages of respiratory viral infections induces systemic metabolic remodeling and that the associated key amino acid catabolites are valid predictors for excessive inflammatory response in later disease stages.

Since late December 2019, the outbreak of a new coronavirus named SARS-CoV-2 has led to the ongoing COVID-19 pandemic[1]. The presentation of COVID-19 can be divided into four groups: asymptomatic infection; infection with mild to moderate symptoms similar to cold or mild pneumonia; infection leading to severe pneumonia and requiring supplemental oxygen or invasive cardiopulmonary resuscitation; and, finally, patients developing long COVID, independent of disease severity[2–4]. Interestingly, some patients with COVID-19 who initially present with a mild form of the disease rapidly develop severe pneumonia, often with fatal outcomes[5]. Previous studies of symptomatic patients have shown that the course of COVID-19 can be divided into various stages. These have been classified as initial infection (stage I),

[1]Center for Cancer Immunotherapy and Immunobiology, Kyoto University Graduate School of Medicine, Kyoto, Japan. [2]Department of Laboratory Medicine, Keio University School of Medicine, Tokyo, Japan. [3]Department of Infectious Diseases, Keio University School of Medicine, Tokyo, Japan. [4]Department of Virology & Parasitology, Graduate School of Medicine, Osaka Metropolitan University, Osaka, Japan. [5]Research Institute for Microbial Diseases, Osaka University, Osaka, Japan. [6]Center for Infectious Disease Education and Research, Osaka University, Osaka, Japan. [7]Faculty of Food and Agricultural Sciences, Fukushima University, Fukushima, Japan. [8]Department of Nephrology, Graduate School of Medicine, Kyoto University, Kyoto, Japan. [9]Institute for the Advanced Study of Human Biology (ASHBi), Kyoto University, Kyoto, Japan. [10]Research Center for Drug Discovery, Faculty of Pharmacy and Graduate School of Pharmaceutical Sciences, Keio University, Tokyo, Japan. [11]Department of Biochemistry, Keio University School of Medicine, Tokyo, Japan. [12]WPI-Bio2Q Research Center, Keio University, and Central Institute for Experimental Medicine and Life Science, Kanagawa, Japan. [13]These authors contributed equally: Rae Maeda, Natsumi Seki. ✉e-mail: yuki.sgi@gmail.com

pneumonia (stage II), and excessive inflammatory response (stage III)[6]. During stage I, from a clinical standpoint, it is difficult to observe signs that would lead to the sudden development of serious pneumonia. To complicate matters further, even patients with severe stage III disease may exhibit a low viral load[7], indicating that the mechanism by which COVID-19 pneumonia becomes severe cannot be simply explained by viral overgrowth at stage III. Unknown virus–host interactions at an early stage may trigger the subsequent maladapted immune responses. This raises the possibility that the severity of late-stage pneumonia could be predicted and traced to changes in the immune system that are detectable during the early stages. Identification of such early predictive markers remains an urgent requirement for the implementation of appropriate therapeutic interventions. Early biomarker discovery studies have focused on the characterization of immunological and biochemical factors in the serum of COVID-19 patients during and after the onset of severe pneumonia[8]. For example, blood samples from patients with severe pneumonia were examined for clinical features[9], and typical blood markers that correlated with severity were identified[10–13]. A comprehensive proteomic and metabolomic exploration of markers using mass spectrometry was also conducted, focusing on the late stages of the disease. Researchers found that levels of protein factors, such as C-reactive protein, metabolites, such as lipid species, and aromatic amino acids, such as tryptophan (Trp), were correlated with disease severity at the time of blood collection[14]. However, while these studies revealed altered molecular signatures as a result of severe pneumonia, there have been few reports concerning prognostic markers in the early stages of infection that would predict rapid deterioration in lung function. More recent studies have reported that patterns of fluctuation in some cytokine levels differ among patient groups with different outcomes[15]. In particular, early-stage levels of IFN-λ and CCL17 have been shown to differ between patients with different outcomes and are suggested to represent candidate prognostic markers[16].

In addition to cytokines, blood metabolite levels may be useful for monitoring inflammatory responses to viral infections and predicting excessive immune response. This is because activated immune cells consume large amounts of metabolites from the blood to enhance immune function and proliferative capacity[17–19]. Studies have shown that serum metabolites were altered in the early stages of SARS-CoV-2 infection[20–22]; however, the underlying mechanisms causing such variation in serum metabolites during the early and late stages of the disease remain unclear. These findings indicate that immune system function can be monitored using the metabolite concentration profile in the blood[18].

The goal of this study was to elucidate the mechanisms of metabolic response during the early stages of the disease to predict disease progression and identify patients at risk of negative outcomes. Furthermore, we aimed to identify early prognostic marker metabolites for tailored therapeutic interventions. To this end, we monitored immune system activity from the early stages of infection (within 5 days of symptom onset) and sought to predict the prognosis using serum metabolome signatures in patients with COVID-19. The metabolite biomarkers, the generation mechanisms of which have been elucidated, serve not only to predict the progression toward severe COVID-19 but also to provide critical insights that could support the development of therapeutic strategies to mitigate disease severity.

## Results
### COVID-19 induces changes in serum metabolite concentrations during early-stage disease that correlate with the severity of future pneumonia

In this study, comprehensive metabolomic and cytokine analyses were performed on serum samples from COVID-19 patients collected from multiple centers (see Fig. 1, Supplementary Table 1, Methods). To determine whether metabolite levels in the early stages of COVID-19 differed among the different outcome groups, we performed a metabolomic analysis of patient sera collected within 5 days of disease onset in Cohort-1, which comprised a total of 71 patients (17, 7, 17, 15 and 15 patients with negative, asymptomatic, mild, and moderate-to-severe outcomes, respectively). We first comprehensively analyzed approximately 300 water-soluble metabolites (Supplementary Data. 1), as well as lipids, including phospholipids (Supplementary Fig. 1 and Supplementary Data. 2), and steroid hormones (Supplementary Fig. 2 and Supplementary Data. 3). We subsequently focused our analysis on water-soluble metabolites, which showed significant differences in the early stages of the disease (Fig. 2). Multivariate analysis of serum metabolites identified independent profiles for each group of patients with different disease outcomes (Fig. 2a), and hierarchical cluster analysis (HCA) revealed the presence of sets of metabolites exhibiting acute changes that correlated with late-stage disease severity (Fig. 2b). We found that patients who developed severe pneumonia in later stages of the disease exhibited enhanced catabolic pathways for various amino acids during the early stages of infection. The most markedly enhanced amino acid catabolic pathway was Trp degradation via the indoleamine 2,3-dioxygenase/ tryptophan 2,3-dioxygenase (IDO/TDO) pathway (Fig. 2c). A decrease in Trp, the substrate, along with an increase in kynurenine, the catabolic product, showed a significant correlation with poor disease prognosis. Further

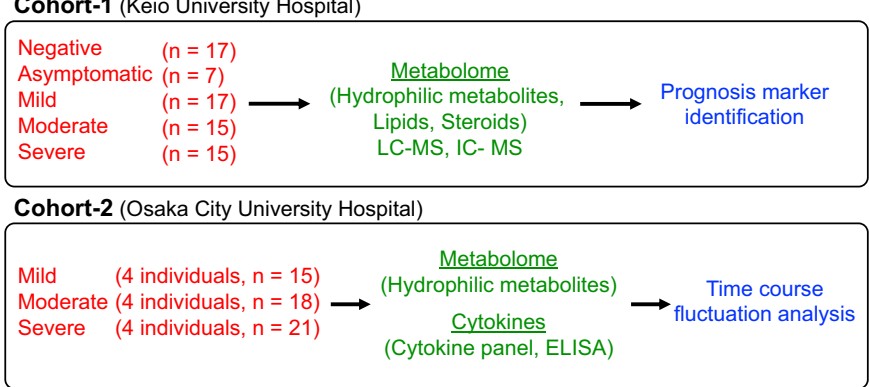

**Fig. 1 | Study design for serum marker discovery, focusing on the early onset of COVID-19.** The Cohort-1 study focused on the analysis of early disease stages (within 5 days of disease onset) and aimed to develop early prognostic markers. Serum samples were used for both metabolomic and cytokine analyses. The Cohort-2 study aimed to elucidate the time specificity of metabolic changes in candidate prognostic markers. For this purpose, blood samples were collected from the same patients at multiple time points, and the temporal changes in blood metabolite levels were examined.

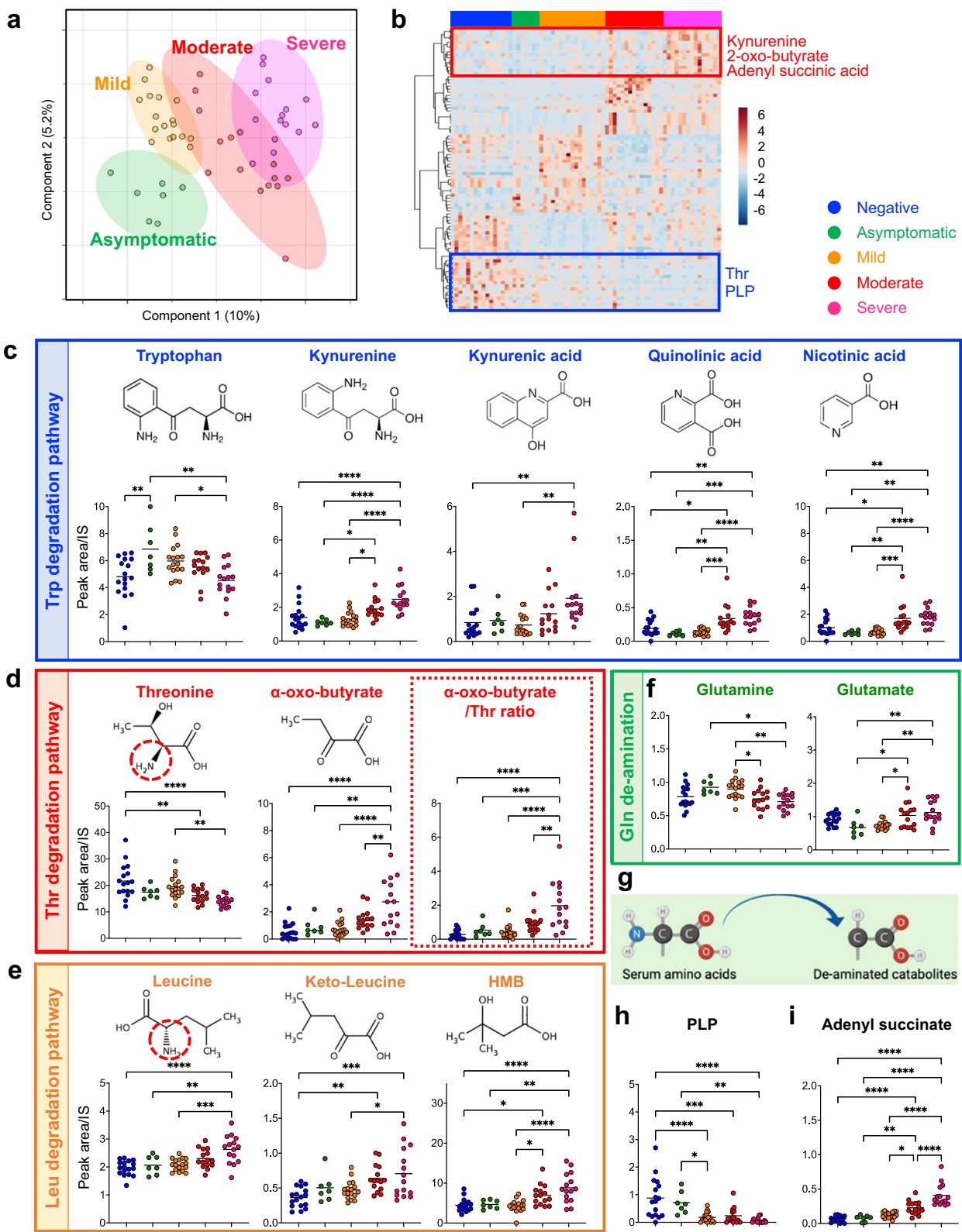

**Fig. 2 | Changes in serum metabolite profile induced during the COVID-19 early-stage disease correlate with future pneumonia severity.** Serum metabolome profiles of Cohort-1 participants are shown. Metabolomic datasets obtained from the sera of SARS-CoV2-negative ($n = 17$), asymptomatic ($n = 7$), mild ($n = 17$), moderate ($n = 15$), and severe ($n = 15$) outcome patients collected within 5 days of disease onset were analyzed using PLS-DA, and independent clusters in each group were located along the X-axis (score 1) in correlation with future severity (**b**). Hierarchical cluster analysis (HCA) was used to classify metabolites with similar patterns of variation and extract metabolites with patterns of increase or decrease that correlated with disease outcome (**a**). Quantification of each metabolite (C–F, H,I). Trp and Trp catabolites via the IDO/TDO pathway (**c**). Thr, 2-oxo-butyrate and 2-oxo-butyrate/Thr (**d**). Leucine, the diamino-catabolite keto-leucine and HMB (**e**). Gln and the diamino-catabolite Glu (**f**). Diagram of deamination (**g**). PLP **h**, and adenyl succinate, which is the intermediate metabolite of the de novo pathway of purine synthesis (**i**). Data are expressed as means. Statistical significance was assessed using one-way ANOVA with Tukey's multiple comparisons test (**c**–**f**, **h**, **i**). * $p < 0.05$; ** $p < 0.01$; *** $p < 0.001$; **** $p < 0.0001$.

downstream, the levels of the deaminated metabolites—kynurenic, quinolinic, and nicotinic acid—were also elevated in the severe group. In addition, threonine (Thr) levels were significantly decreased, while those of 2-oxo-butyrate, a catabolite produced by the deamination of Thr, increased in a complementary manner, showing a significant difference even between the moderate and severe groups (Fig. 2d). Although the blood levels of leucine (Leu), a branched chain amino acid (BCAA), increased, the concentrations of both keto-leucine, a deaminated product, and downstream hydroxy-β-methylbutyrate (HMB) increased; both of these changes were correlated with poor prognosis (Fig. 2e). Notably, the levels of glutamine (Gln), the amino acid with the highest blood concentration[23], decreased, while those of its deaminated product, glutamic acid (Glu), increased (Fig. 2f) in the moderate and severe groups compared to the mild or asymptomatic groups. The levels of other amino acids including histidine (His) decreased in patients with poor prognosis, and those of urocanic acid, a product of His deamination, were significantly higher in the moderate and severe groups than in the other groups (Supplementary Fig. 3b). These results indicate the accelerated deamination of amino acids (Fig. 2g). Interestingly, levels of pyridoxal 5′-phosphate (PLP), which acts as a cofactor for aminotransferase to catalyze deamination, decreased markedly with worsening prognosis (Fig. 2h), systemically promoting the aminotransferase reaction.

The enhancement of amino acid degradation pathways is often observed in the serum of lung cancer patients[24] and is involved in accelerated de novo nucleic acid synthesis[25]. Notably, levels of adenyl succinate, an intermediate metabolite of the de novo synthetic pathway of purine nucleotides, were elevated in patients with poor prognosis (Fig. 2i). Moreover, adenosine monophosphate (AMP) was detected only in the sera of COVID-19-positive patients, and the frequency of detection increased with worsening prognosis, with a detection rate of 100% in patients with severe outcomes (Supplementary Fig. 3c). Conversely, levels of hypoxanthine, a purine degradation product, were lower in the severe group than in the mild group (Supplementary Fig. 3d). It should also be noted that fluctuations in bioactive molecules in the blood, including increases or decreases in steroid hormones (Supplementary Fig. 2), were observed from the early onset of COVID-19. Patients with a severe course of COVID-19 have previously been reported to have increased cortisol[26] and decreased testosterone[27] levels from early infection onward, a trend consistent with that observed in this study.

## Serum amino acid catabolite levels act as an early marker for predicting COVID-19 outcome by reflecting initial viral load

We found that several serum amino acids and their catabolite levels were significantly correlated with PCR-Ct values or SARS-CoV-2 viral load (Fig. 3a–f, Supplementary Table 2). For example, levels of Trp and its catabolites 3-hydroxykynurenine and kynurenine were inversely correlated with PCR-Ct values (Fig. 3a–d). The decrease in Thr levels and increase in Glu levels were also significantly correlated with PCR-Ct values, suggesting that the catabolism of these amino acids was correlated with viral load during the early stages of infection (Fig. 3e, f).

Receiver operating characteristic (ROC) curves of blood metabolites measured at the early disease stage were then analyzed to evaluate their efficacy as prognostic markers for discriminating between severe and moderate (Fig. 3g), severe and mild (Fig. 3h), and severe and asymptomatic (Fig. 3i) disease outcomes. The results showed that several amino acids and their deaminated catabolites were effective prognostic markers in all comparisons (Supplementary Table 3, Fig. 3g–i). Serum kynurenine levels were useful as an early marker for discriminating between severe and moderate (AUC = 0.786), severe and mild (AUC = 0.982), and severe and asymptomatic (AUC = 1.00) disease (Fig. 3, Supplementary Table 3). Adenyl succinate, the intermediate metabolite of the purine nucleotide synthesis pathway, showed the best accuracy for differentiating all groups (AUC =

0.845 for severe vs. moderate comparison). Furthermore, the levels of serum metabolites (including the aforementioned amino acid catabolites) and adenyl succinate could help differentiate patients with mild disease and asymptomatic patients with approximately 100% accuracy in the early stages of disease onset.

We also examined whether serum metabolites correlated with COVID-19 prognosis were affected by the presence and type of comorbidities (Supplementary Fig. 4), surface symptoms (Supplementary Fig. 5) or by future treatment (i.e., oxygenation or airway intubation) (Supplementary Fig. 6). While several metabolites were affected in this manner, this was not the case for any of the amino acids and their catabolites discussed here (See detail for the Supplementary Data).

## Increased amino acid deamination is specific to early-stage disease in patients progressing to severe disease

To test whether the early metabolic changes that correlated with disease severity were specific to only early stages of disease, we analyzed sequential samples from Cohort-2, comprising a total of 12 patients (4 each with mild, moderate and severe outcomes, respectively). The results of HCA, shown in Fig. 4a, indicate that serum levels of metabolite groups including amino acids and associated catabolites were significantly elevated (Cluster-i) or reduced (Cluster-ii) with the progression of pulmonary inflammation.

Subsequent analysis revealed that enhanced amino acid catabolism was specific to the early disease stage in patients with poor prognosis; the decreased levels of amino acids (Trp, Thr, and Gln) and increased levels of corresponding diamino catabolites were found to be specific to the early stages (Fig. 4b–l). Trp levels decreased markedly from the early to late stages in the severe group, only to increase slightly toward the late stage (Fig. 4b, c). By contrast, the levels of Trp degradation metabolites, including kynurenine (Fig. 4d) and quinolinic acid (Supplementary Fig. 7a), showed a consistently increasing trend from early to late disease stages. We also found that the decrease in Thr (Fig. 4e) and the production of its deaminated catabolite, 2-oxo-butyrate (Fig. 4f, g), were more pronounced in the early phase and tended to recover in the late phase. Similar trends were observed for Gln (Fig. 4h, Supplementary Fig. 7g) and its deaminated product Glu (Fig. 4i, Supplementary Fig. 7h). Although blood levels of leucine did not change (Fig. 4J), the elevated levels of HMB (Fig. 4k, l), b-hydroxy-iso-butyrate (Fig. 4m), and acetoacetate (Fig. 4n) resulting from BCAA deamination were also specific to the early stages of the disease. Notably, serum AMP levels consistently increased from the early stage of the disease in patients with severe outcomes (Supplementary Fig. 7i). Conversely, levels of hypoxanthine, uric acid, and allantoin, which are purine degradation products, were markedly reduced (Supplementary Fig. 7j, k).

We observed an increase in the levels of serum metabolites that appeared to be deviated metabolites from specific organs; for example, levels of creatine, which is thought to be produced in the liver and transferred to muscle tissue[28], were elevated in the late stage of the disease (Fig. 4o). Interestingly, the levels of neurotransmitters such as neuropeptide N-acetylaspartylglutamate (NAAG) and melatonin were also elevated (Fig. 4p, q), suggesting leakage from the central nervous system associated with blood–brain barrier damage.

## Combined serum cytokine and metabolite levels predict COVID-19 severity outcomes more accurately than do cytokine levels alone

As previously reported, the early elevation of certain blood cytokine levels can act as a prognostic predictor of COVID-19, reflecting an increased risk of developing severe pneumonia[29]. Therefore, we quantified a serum metabolome and cytokine profile from the same samples to evaluate which was a better predictor of prognosis. In Cohort-1 samples, eight cytokines were found to be specifically

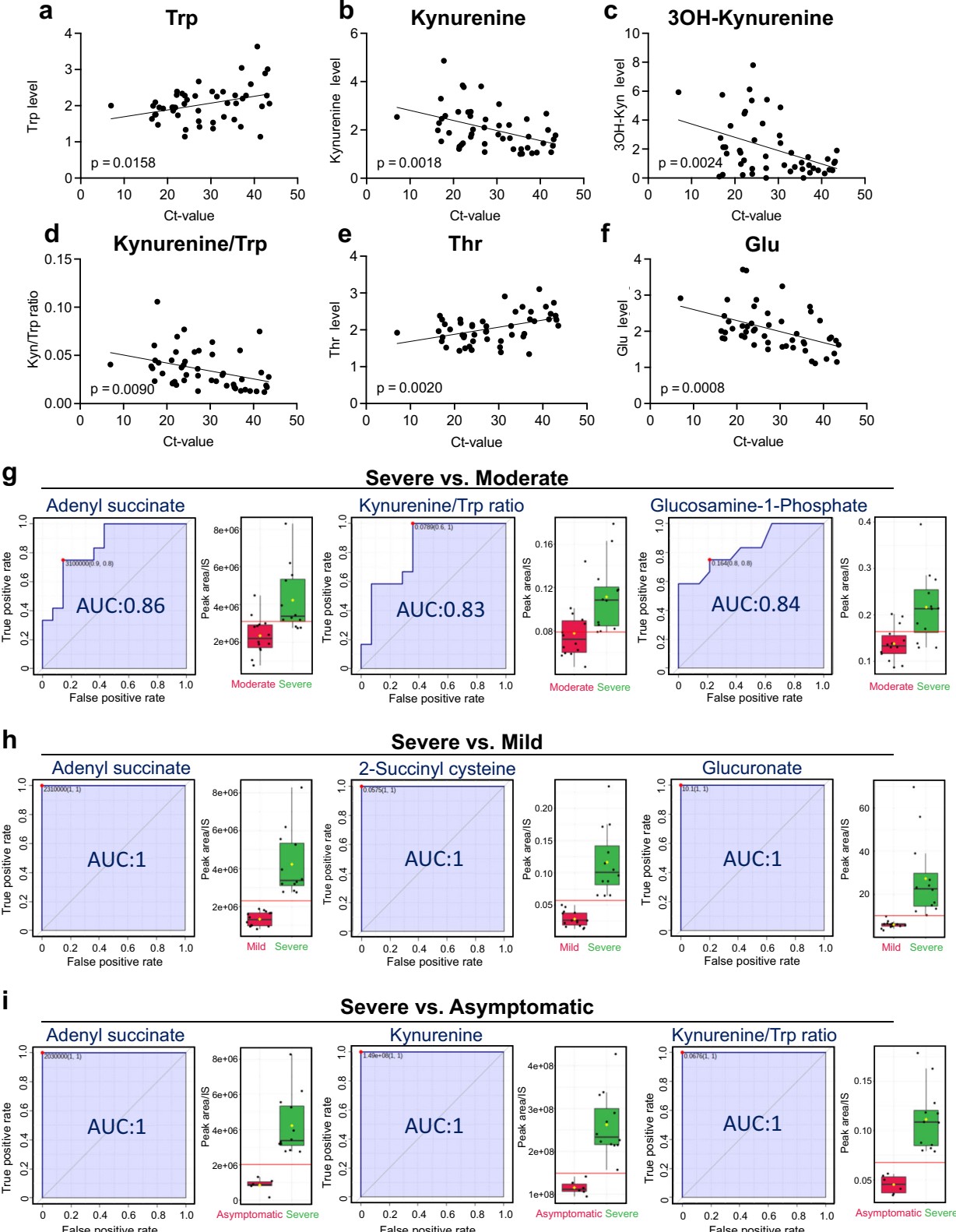

**Fig. 3 | Specific serum amino acid levels and their catabolite levels are prognostic predictors reflecting viral load during early COVID-19.** Using samples from early COVID-19 onset (Cohort-1 samples), correlation analysis between the metabolite levels (peak area/IS) and PCR-Ct values was performed. Two-sided Pearson correlation (r) and *p*-values are indicated (**a**–**f** and Supplementary Table 2). Metabolites that could potentially discriminate COVID-19 severe pneumonia patients (*n* = 15) from moderate (*n* = 15), mild, and asymptomatic (*n* = 7) groups were evaluated using the ROC curve analysis (**g**–**i**). The central line in the box plot represents the median, the upper limit corresponds to the 75th percentile, and the lower limit corresponds to the 25th percentile. The whiskers extend to the maximum and minimum values.

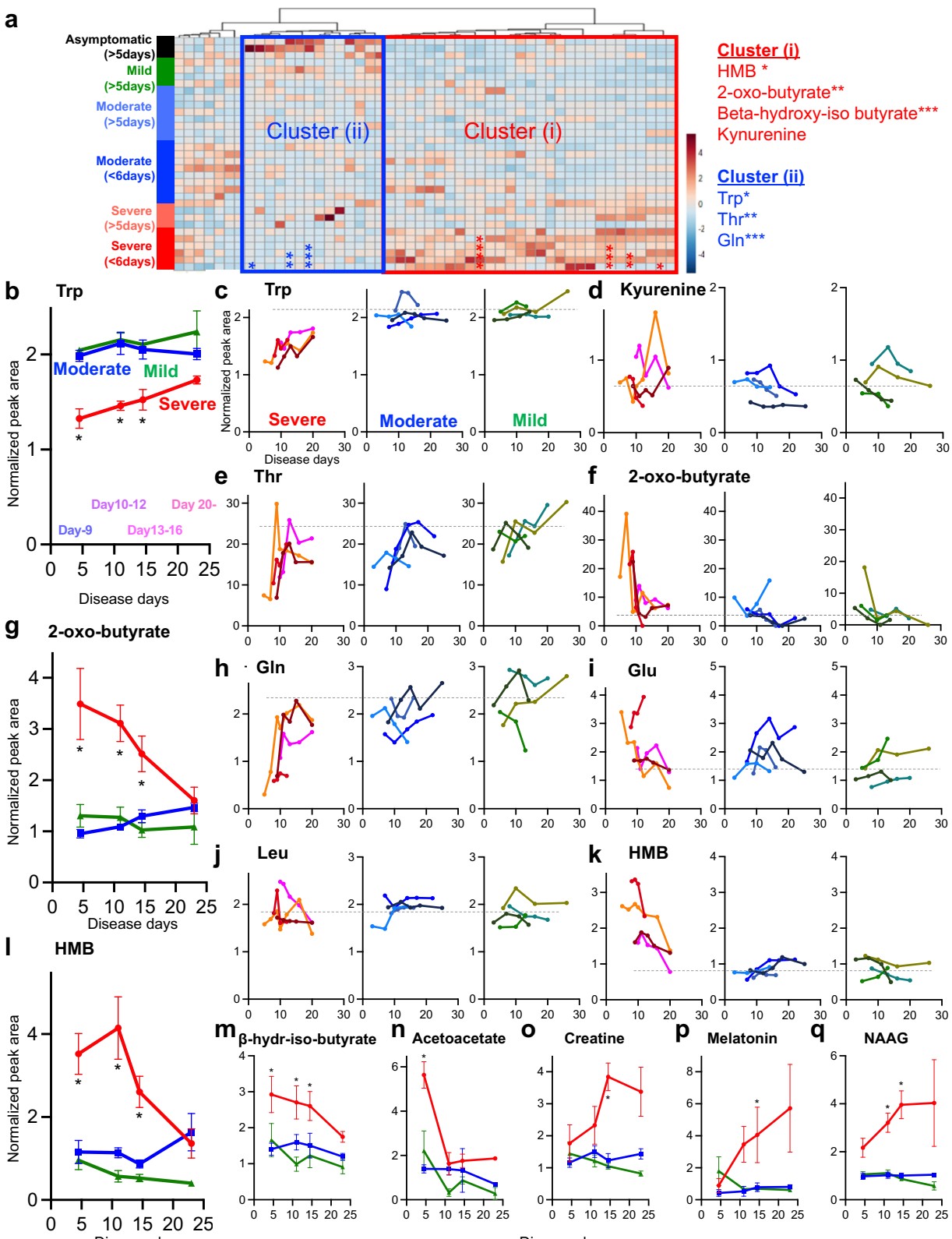

**Fig. 4 | Increase in serum amino acid diamino catabolism was specific to the early stage of COVID-19.** Serum metabolome profiles of Cohort-2 participants are shown. In hierarchical cluster analysis, the presence of metabolite clusters that increased (Cluster-i) or decreased (Cluster-ii) with disease severity is indicated (**a**). Fluctuations in serum metabolite levels were plotted against the days of disease for each patient or group with different COVID-19 outcomes (**b**–**q**). Time course of serum metabolite levels at each of four time points: day 9, days 10–12, days 13–16, and day 20 from disease onset for the severe, moderate, and mild disease patient groups (**b**, **g**, **l**, **m**–**q**) (*n* = 2–4). Serum metabolite levels were plotted for individual patients with severe, moderate, and mild disease as they changed with day of illness (four patients each. Severe, 21 samples total; moderate, 15 samples total; mild, 15 samples total) (**c**–**f**, **h**–**k**). Data are expressed as mean value ± SEM. Statistical significance was assessed using two-way ANOVA with Tukey's multiple comparisons test (**b**, **g**, **l**, **m**–**q**). * *p* < 0.05.

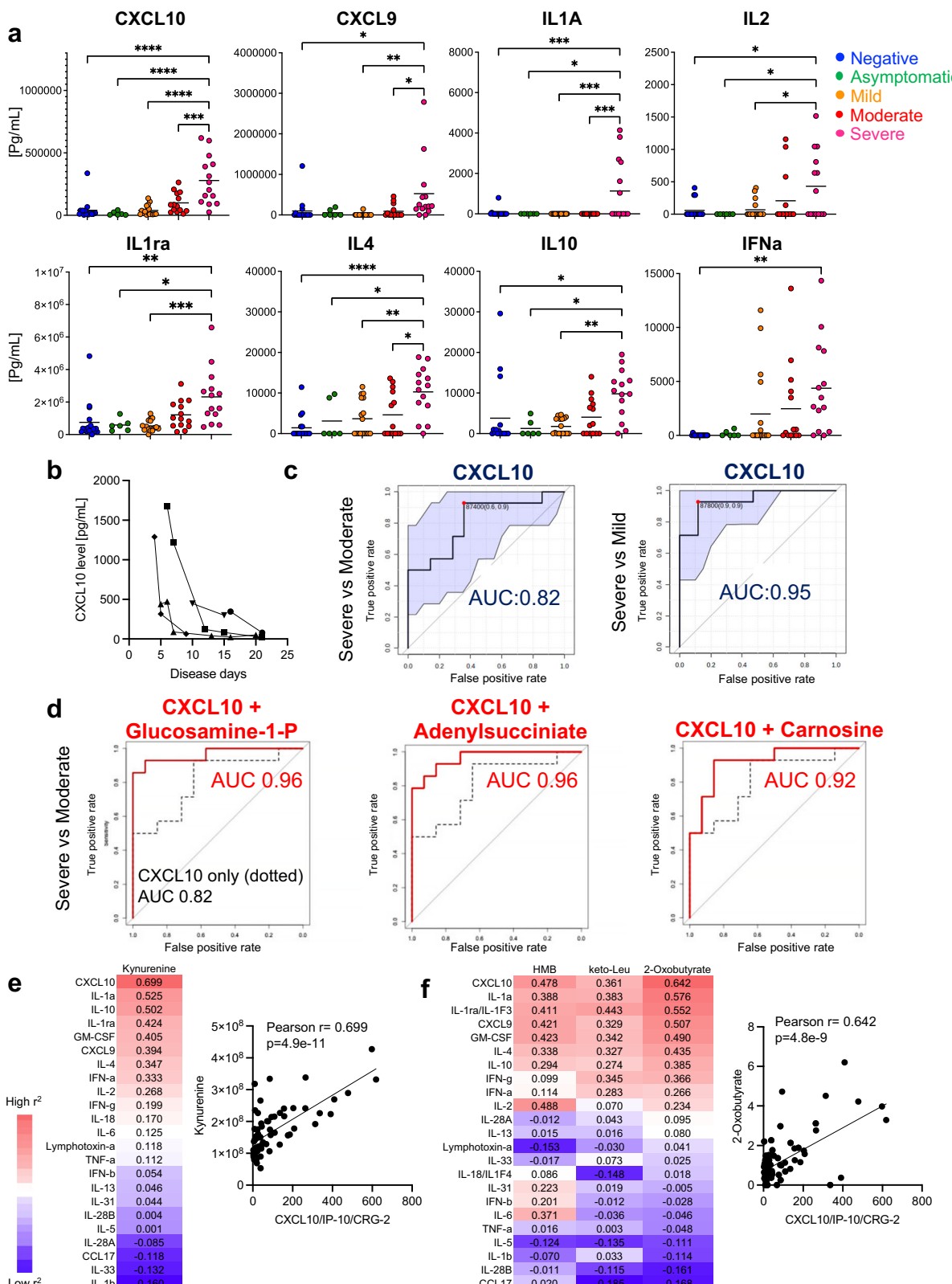

elevated during the early stages of the disease in patients with severe outcomes (Fig. 5a, Supplementary Table 4). Of these, blood CXCL10 levels were markedly elevated and correlated with worse outcome. Validation in the Cohort-2 sample group showed that the CXCL10 elevation was specific to early-stage disease (Fig. 5b).

Next, ROC curve analysis was used to evaluate the prognostic performance of blood cytokine levels, and CXCL10 showed the best predictive performance (Supplementary Table 4, Supplementary Figs. 8–10). The AUC was 0.824 for severe vs. moderate outcome discrimination and 0.95 for severe vs. mild group discrimination (Fig. 5c). Notably, the prognostic performance (AUC values) of CXCL10 alone was lower than that of the combined serum metabolites. Therefore, we examined whether combining the blood metabolite and cytokine (CXCL10) profiles could improve prognosis performance. A

**Fig. 5 | Serum CXCL10 levels in combination with specific metabolites predict COVID-19 severity more accurately than CXCL10 alone.** Multiple cytokines were simultaneously quantified by cytokine panel analysis in parallel with metabolomic quantification using the same samples from the early stages of COVID-19 onset (Cohort-1 samples). Twenty-three cytokines were quantified, and those that were significantly elevated in the severe group were compared to those in the other groups (CXCL10, CXCL9, IL2, IFN-a, IL1ra, IL4 and IL10) (**a**). Time-course analysis of CXCL10 level of Cohort-2 patients with severe outcomes (**b**). Each form represents one individual. ROC curve analysis was performed to evaluate the discriminative power of CXCL10 in severe vs. mild and severe vs. moderate groups (**c**). ROC analysis of combination of serum CXCL10 and metabolites (glucoasamine-1-phosphate, adenyl succinate, and carnosine) of severe vs. moderate groups (**d**). The two-sided Pearson's correlation between cytokines and serum kynurenine concentrations (left panel), and a correlation plot between kynurenine and CXCL10 (right panel) (**e**). Correlations with cytokines were similarly evaluated for HMB and keto-Leu (deaminocatabolites of Leu) and 2-oxo-butyrate (a deaminocatabolite of Thr), and results are presented as in (**e**, **f**). Data are expressed as means. Statistical significance was assessed using one-way ANOVA with Tukey's multiple comparisons test (**a**). * $p < 0.05$; ** $p < 0.01$; *** $p < 0.001$; **** $p < 0.0001$.

comprehensive analysis of metabolites and cytokine synthetic variables shows that the combination of CXCL10 with glucosamine-1-phosphate, adenyl succinic acid, and carnosine improved the predictive performance and the AUC of CXCL10 alone [0.964, 0.9643, and 0.9184, respectively, compared to CXCL10 alone (0.824)] (Fig. 5d). Conversely, the combination of CXCL10 with kynurenine did not improve prognostic performance (Supplementary Fig. 11). A comprehensive examination of the correlation between metabolites and cytokines showed that the intermediate metabolites of the IDO pathway, namely, kynurenine, 3-hydroxykynurenine, quinolinic acid, nicotinic acid, and BCAA, and their catabolites form a cluster that is highly correlated with CXCL10 (Supplementary Fig. 12; red square). Among all cytokine-metabolite combinations, the highest positive correlation was observed for kynurenine and CXCL10 (Fig. 5e), suggesting that CXCL10 and kynurenine may share the same signaling pathway. In addition, deaminated metabolites of Thr and BCAA showed a strong positive correlation with CXCL10 (Fig. 5f), suggesting that these amino acid catabolic pathways are regulated by the same signal as CXCL10, possibly the IFNγ-CXCL10/9 axis.

## Metabolic remodeling of mouse lungs during early SARS-CoV2-MA10 and influenza infection leads to elevated de novo nucleotide synthesis

It is reasonable to assume that the infected lung tissue is the primary source of metabolite changes in the serum of patients with early COVID-19. Thus, the increase in the diamino-catabolic pathway activity of amino acids seen in patient serum may have been the result of an increase in the de novo nucleic acid synthesis pathway associated with the proliferation of certain cells in the lungs. To test this hypothesis, we collected lung tissue from mouse models of early infection (3 d after infection) with two respiratory RNA viruses, mouse susceptible SARS-CoV2-MA10[8,30] and influenza virus and performed metabolomic analysis. When C57BL/6 J (B6, mild disease model) and Balb/c (severe disease model) mice were infected with SARS-CoV2-MA10, more severe weight loss was observed in the Balb/c strain (Fig. 6a). Using these two models, we examined the correlations between disease severity, lung metabolomic remodeling, and cytokine production. Figure 6b shows that mRNA expression of *Ifng, Cxcl10*, and *Cxcl9* was significantly elevated in the severely affected Balb/c mice, while the B6 mice showed a lower degree of elevated expression or no significant difference. There was no significant increase in *IL6* expression in either strain. Among amino acid-metabolizing enzymes, *Ido1, Ido2*, and BCAA transaminase 1 (*Bcat1*) were strongly upregulated in the lungs of infected Balb/c mice. In addition, the expression of *Ki67*, a marker of cell proliferation, was elevated, suggesting that some type of cell proliferation was enhanced. These results were common not only in the SARS-CoV2-MA10 infection model but also in the early influenza infection model, indicating that this is an early response common to respiratory RNA viruses.

In addition, lung metabolomic analysis revealed enhanced amino acid catabolism and de novo nucleotide synthesis pathway activity in the severely affected Balb/c mice compared to that in B6 mice; the production of formyl-kynurenine, kynurenine, and 3-hydroxykynurenine was significantly elevated, as was that of the downstream product quinolinic acid, required for de novo NAD synthesis (Fig. 6c). These observations are suggestive of enhanced IDO pathway activity. In addition, levels of branched chain keto acids (BCKAs), deaminated catabolites of BCAA, were elevated only in the severely affected mice (Fig. 6d). Interestingly, in the Balb/c mice, signs of elevated disease severity (i.e., weight loss) exhibited accelerated progression with higher viral exposure, and the concentration of BCKA and its downstream BCAA catabolites increased significantly with disease severity (Fig. 6e). Furthermore, the deamination reactions of Thr and Glu were promoted as the levels of deaminated catabolites of Thr (2-oxo-butyrate) and Gln (Glu) increased (Fig. 6f). Conversely, the levels of substrates such as Trp, BCAA, and Thr appeared to remain constant or increase in the lung tissues, possibly due to increased demand of the infected tissues (Fig. 6c–f). The levels of metabolites of the purine de novo synthesis pathway derived from phosphoribosyl pyrophosphate (PRPP) and of purine nucleotides also increased, indicating a marked increase in the de novo synthesis of purines in the lungs (Fig. 6g). Similarly, production of metabolites of the pyrimidine de novo synthesis pathway, namely, pyrimidine nucleotides, was elevated. In particular, elevated thymidine nucleotide production suggested that DNA synthesis, i.e., nucleic acid synthesis necessary for cell proliferation, was enhanced (Fig. 6h). Similar metabolic remodeling in the lungs was observed in the influenza virus infection model (Supplementary Fig. 13).

## Visualization of nucleotide accumulation in hyper-proliferated vascular smooth muscle and airway epithelial cells of SARS-CoV2-MA10- and influenza-infected mouse lungs

H&E staining of infected lung tissues revealed abnormal proliferation of mucosal secretory cells in bronchial tissue and smooth muscle in vascular tissue in both SARS-CoV2-MA10 and influenza mouse models (Fig. 7a). Histopathological examination of the lungs showed both thickening of the vascular smooth muscle (red arrows) and hyperproliferation of the airway mucosal ciliated secretory cells (blue arrows) in the SARS-CoV2-MA10-infected lungs. This trend was even more pronounced in the influenza infection model (Fig. 7a, b). The thickened mucosal ciliated secretory cells were Periodic acid-Schiff (PAS)-positive, suggesting active mucus secretion (Fig. 7b; bottom). In such remodeled airway tissue, SARS-CoV-2 spike protein was detected in type 2 epithelial cells (Fig. 7c). Interestingly, anti-CXCL10 antibodies strongly stained epithelial cells in a pattern that merged with this SARS-CoV-2 signal, suggesting CXCL10 production by infected airway epithelial cells (Fig. 7d). Because proliferating cells in the pulmonary airways and blood vessels remodeled by viral infection were predicted to be responsible for increased de novo nucleotide synthesis, we tested this hypothesis by mapping metabolites in lung tissue via imaging mass spectrometry (IMS). Notably, levels of inosine monophosphate (IMP), an intermediate metabolite of adenine nucleotides, were markedly elevated in thickened vascular smooth muscle in both Balb/c and B6 mice (Fig. 7e, f; red arrows). In addition, adenyl succinate was strongly detected in thickened airway epithelial cells, suggesting that purine synthesis was also active in these cells (Fig. 7f, blue arrows).

Because the lungs of SARS-CoV2-MA10-infected mice were thinly sectioned after tissue heat treatment to inactivate the virus due to biosafety concerns, high-energy metabolites (e.g., ATP)

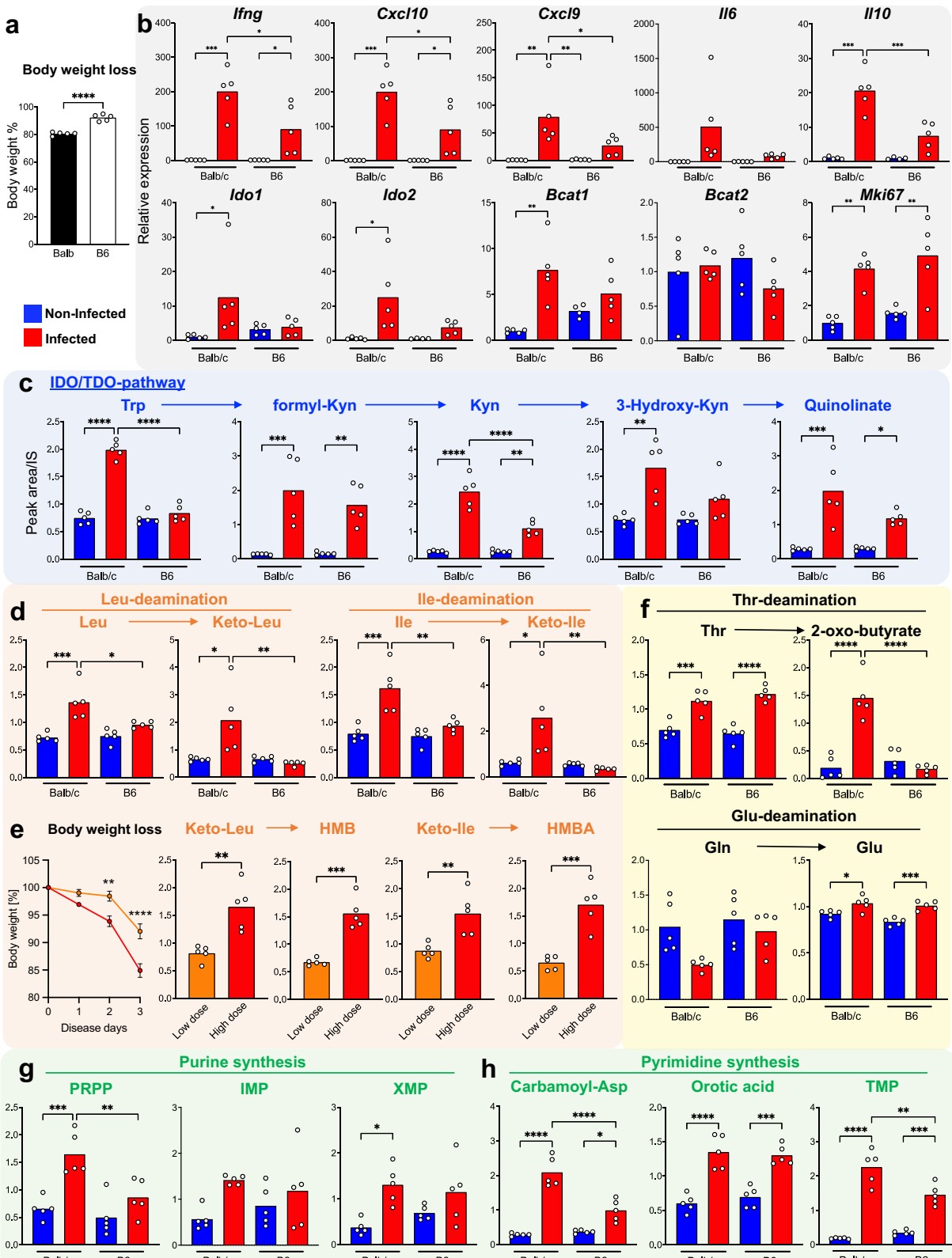

**Fig. 6 | Metabolic remodeling with progressive deamination of amino acids and production of de novo nucleotides occurs in lung tissue during early SARS-CoV2-MA10 infection.** Balb/c and C57BL/6 (B6) strain mice were infected with SARS-CoV2-MA10 (*n* = 5). Body weight change from original mouse weight (%) on day 3 of infection (**a**). The mRNA expression of cytokines, chemokines, metabolic enzymes, and *Mki67* (as a cell proliferation marker) were quantitatively evaluated using qPCR on lung tissues of the SARS-CoV2-MA10-infected mice on day 3 of infection and from uninfected Balb/c and B6 mice (**b**). Metabolomic analysis was performed using lung tissue of the two strains on day 3 of infection (**c**–**h**) to quantitatively evaluate metabolites of the IDO/TDO pathway (**c**), BCAA and its catabolites (**d**), Thr and Gln and their catabolites (**f**), and metabolites of the de novo nucleotide synthesis pathway (**g**, **h**). Within the Balb/c strain, we quantitatively evaluated weight loss and BCAA catabolite production during low and high doses of viral infection (**e**). Data are expressed as mean value ± SEM. Statistical significance was assessed using one-way ANOVA with Tukey's multiple comparisons test. * *p* < 0.05; ** *p* < 0.01; *** *p* < 0.001; **** *p* < 0.0001.

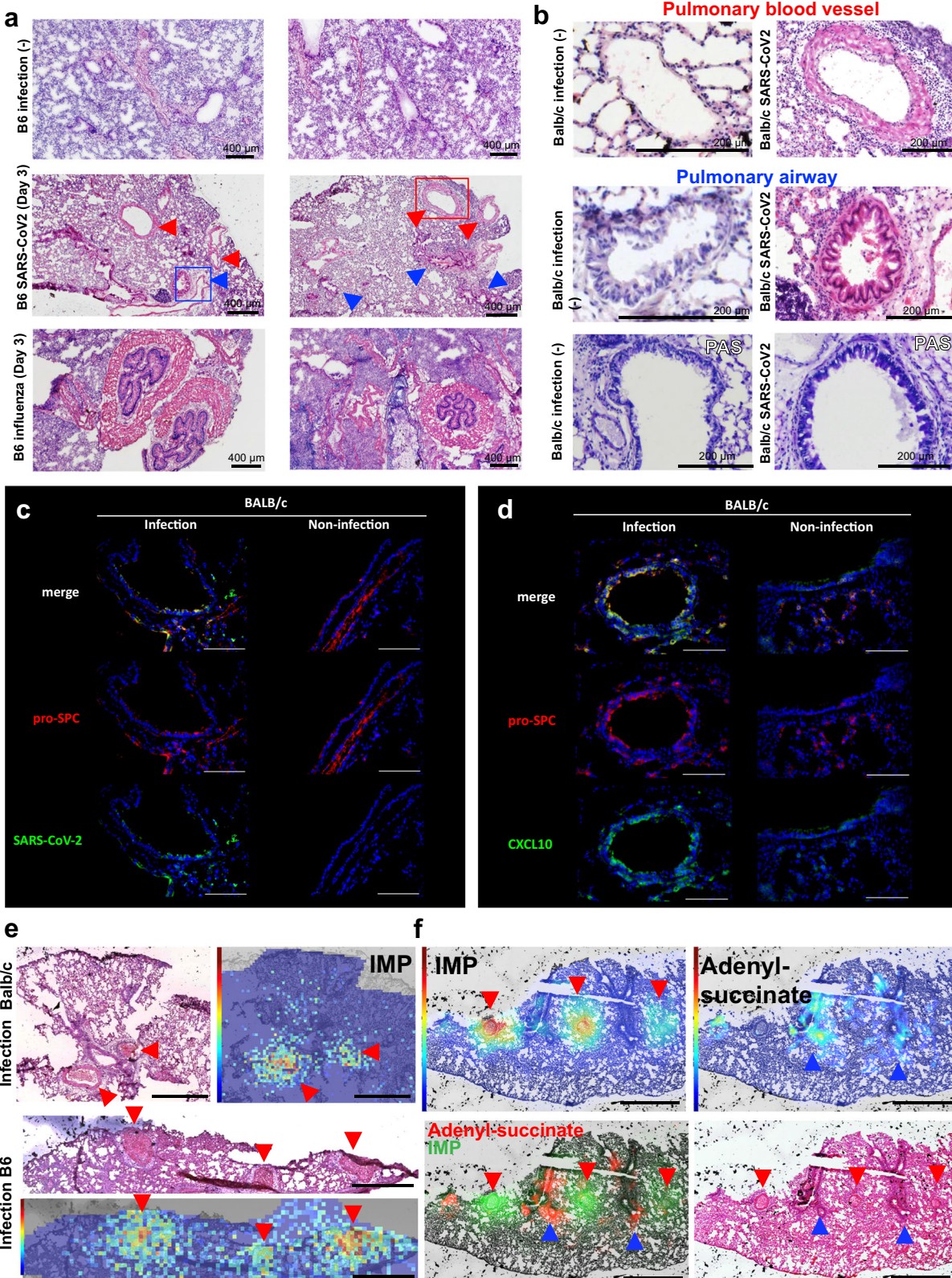

**Fig. 7 | In SARS-CoV2-MA10- and influenza-infected mouse lungs, nucleotide accumulation was enhanced in hyperproliferative vascular smooth muscle and airway epithelial cells.** The pathogenesis of early infection was compared by H&E staining in lung tissue from B6 strain mice infected with SARS-CoV2-MA10 and influenza. Hyperproliferation of vascular smooth muscle (red) and mucosal secretory cells (blue) are shown with arrows (**a**). In the SARS-CoV2-MA10 infection model lung, the area enclosed by the red and blue squares in panel A was enlarged to show proliferated vascular smooth muscle and airway epithelial cells (**b**), and

PAS staining was performed (b, bottom). Fluorescence image of immunohistochemical staining with anti-Pro-SPC and anti-SARS-CoV-2 spike protein antibodies (**c**), and staining with anti-Pro-SPC and anti-CXCL10 spike protein antibodies (**d**) in SARS-CoV2-MA10 infected airway tissue. Visualization of IMP, a representative metabolite of de novo nucleotide synthesis, by imaging MS revealed significant accumulation in the expanded smooth muscle tissue (red arrow) (**e**). Adenyl succinate accumulated in remodeled airway tissue (blue arrow) (**f**). Data are representative of three independent experiments (**a**–**f**).

could not be readily detected via imaging. Therefore, we mapped more nucleotides in influenza model lungs, for which fresh-frozen tissue was available for analysis. As a result, IMP was strongly detected in the hyperproliferative smooth muscle (Supplementary Fig. 15b; red arrows). In addition, ATP was strongly generated in the thickened airway epithelial cells (Supplementary Fig. 15b; blue arrows). Thus, remodeling of the lung parenchyma with specific cell proliferation characteristic of early respiratory virus infection may have been responsible for the metabolic changes associated with the observed increase in nucleotide synthesis. Taken together, our results indicate that the initial response during viral infection is characterized by a high proliferation of the mucosal secretory cells and smooth muscle that make up the airways, accompanied by an increase in de novo nucleotide synthesis.

## Discussion

To date, the immunopathological mechanisms in the early stages of severe pneumonia are not fully understood, and the identification of prognostic markers based on such mechanisms remains a challenge. In this study, we found that from the early stages of SARS-CoV-2 infection (within 5 days of onset), the composition of serum metabolites showed changes that correlated with future disease severity (Fig. 2), and that a combination of cytokine and metabolite levels in the blood could provide a highly accurate prognostic estimate (Fig. 5). Metabolic remodeling associated with the proliferation of cells constituting the pulmonary airways and vasculature can be suggested as a mechanism for this systemic metabolic shift (Figs. 6 and 7).

Cytokine screening analysis performed simultaneously with metabolomic analysis revealed that CXCL10 levels were specifically elevated and significantly positively correlated with blood amino acid catabolite concentrations during the early stages of COVID-19 in severe outcome patients (Fig. 5a, e, f). A simple interpretation of this positive correlation is that the production of IFN-γ, a cytokine which is known to induce the expression of both CXCL10 and the metabolic enzyme IDO[31], by the immune system correlates with the level of viral infection early in the disease, acting on immune cells or lung parenchymal cells to induce CXCL10 and IDO, possibly along with other metabolic enzymes. The fact that both CXCL10 production and amino acid catabolism were reduced in the late phase of the disease (Fig. 4), when viral load was reduced[32], also supports this hypothesis. Very recently, an increased KYN/TRY ratio due to IDO has also been reported in patients with severe COVID[33]. Since IDO and its catabolites kynurenines suppress T-cell responses and promote Treg proliferation[34,35] this may be a strategy of the virus to induce immune tolerance. An important result of this study is that the intensity of the initial infection response to SARS-CoV-2 infection may be quantitatively assessed by the levels of both CXCL10 and amino acid catabolites. Although a clinical examination with comprehensive metabolite measurements is not feasible, an integrated evaluation of these few key metabolites and cytokines (especially CXCL10) in SARS-CoV-2-positive patients would allow for a practical and accurate risk assessment of the potential for severe outcomes in the early stages of infection.

Interestingly, the footprints of the altered metabolites in the serum of COVID-19 patients identified in this study were similar to those in the blood and tumor tissue of lung cancer patients[36]. For example, IDO is strongly expressed in non-small cell lung cancer, while Trp and kynurenine levels are decreased and increased, respectively, in the lungs of patients, with corresponding changes in the blood levels of these compounds[37,38]. Increased deamination of other amino acids is another metabolic footprint commonly seen in lung cancer[39]; deamination of BCAA by BCAT is enhanced in non-small cell lung cancer[25,40] and associated with de novo nucleotide synthesis, which utilizes BCAA-derived amino groups[25]. Moreover, Gln is catabolized by active deamination reactions (i.e., glutaminolysis) in proliferating cancer cells, and the Gln-derived amino group is also used for de novo nucleotide

synthesis[41]. Furthermore, PLP is a coenzyme commonly used in this deamination reaction and was found to be a frequently depleted metabolite in the serum of lung cancer patients[42,43]. Importantly, the metabolic footprint similarities between cancer and COVID-19 patients indicate that amino acid-derived amino groups are actively utilized in the de novo synthesis of nucleic acids, which are necessary for active cell proliferation[24]. The enhanced catabolism of Trp by the IDO pathway may also partially contribute to the synthesis of NAD. Thus, it is possible that active cell proliferation with de novo nucleic acid synthesis occurs in lung parenchymal cells rather than immune cells during early COVID-19 with risk of poor prognosis. Important genes involved in the proliferation of lung parenchymal cells, such as *Karas*, *Akt*, and *Igf*, have been reported as commonly upregulated genes in the lung tissues of COVID-19 and influenza patients[43]. Furthermore, our data indicate that this metabolic event with cell proliferation is specific to early COVID-19 (Fig. 4). Thus, the increased amino acid catabolism corresponding to early viral infection levels probably contributes to the abnormal proliferation of lung parenchymal cells (airway epithelial and vascular endothelial cells; also see below); importantly, a greater magnitude of this cell proliferation corresponds to more severe late-stage pneumonia.

Because the SARS-CoV-2 infection experiments conducted in this study were performed in a biosafety level 3 (BSL-3) environment, fresh-frozen tissue could not be taken out of the facility, and specimens that had been inactivated by heat treatment were used[44]. Therefore, the detection of enzymatically labile metabolites such as ATP was not possible. However, we were able to overcome this limitation to some degree by performing metabolome analysis and IMS using frozen specimens from influenza virus infection experiments. The observed histological changes were similar to the initial pathogenesis of conditions associated with chronic pneumonia, such as asthma, which is triggered by inflammation by cells in the innate immune system, including neutrophils and macrophages[45,46]. Interestingly, such thickening of airways and pulmonary vessels seemed to disappear in the tissue of SARS-CoV-2 and late-stage influenza-infected lung models. This finding suggests that this tissue remodeling is specific to the early stages of SARS-CoV2-MA10 and influenza infection and that as the disease progresses, these airway component cells proliferate and release accumulated nucleotides due to necrosis. Since high levels of accumulated nucleotides are thought to promote immune cell activation (e.g., damage associated molecular patterns), it is reasonable to assume that the active synthesis of nucleotides during the early stages of infection would result in more pronounced lung tissue damage during the late stages of infection. Indeed, differences in susceptibility to SARS-CoV2-MA10 and late mortality between Balb/c and B6 strains have been reported[47], and our finding that more prominent de novo nucleotide synthesis from amino acids was enhanced in the lungs of more severely affected Balb/c strain mice supports the above hypothesis (Fig. 6). Furthermore, since BCAA diamino catabolism was specific to severe illness and increased in response to increased viral exposure (Fig. 6e), BCAA catabolites represent a metabolite prognostic marker that accurately predicts the intensity of the initial metabolic remodeling response leading to severe pneumonia.

There are several important reasons to further investigate the relationship between sex differences in human systemic amino acid metabolism and differences in the severity of clinical manifestations of COVID-19. First, it is apparent that amino acid profiles differ between men and women; in particular, the blood levels of BCAAs and subsequent catabolites are higher in men[48]. Furthermore, the existing literature emphasizes that BCAA metabolism is enhanced in men with COVID-19 compared with that in women; moreover, this pattern is similar for tyrosine and tryptophan catabolism[49]. Our study also showed that increased catabolism of these amino acids correlates with poor prognosis (Fig. 2). The mechanism underlying this phenomenon is suggested to be the hyperproliferative response of lung

parenchymal cells in the early stages of infection (Fig. 7). Although a direct causal relationship remains speculative, the increased metabolic reactivity observed in the lung parenchyma of male participants, particularly increased amino acid catabolism, may therefore contribute to the increased male mortality associated with COVID-19[50]. A major limitation of our study is the predominance of men in the moderate and severe groups, which limited our ability to perform comprehensive sex-specific analyses. This distribution makes rigorous statistical investigation of sex differences difficult. Cohort-1 had one female patient in the severe category and one in the moderate category, while Cohort-2 had no females in the severe, one in the moderate, and two in the mild categories out of 4 patients for each group. This distribution makes a robust statistical evaluation of sex differences difficult. In particular, an analysis of Cohort-2 (comprising 12 patients) aimed at tracking the temporal progression of sex differences yielded inconclusive results due to sample size limitations. The implementation of larger studies that account for sex differences is paramount to filling this knowledge gap.

Another limitation of this study is that lung tissue events were only observed in a mouse model. In future studies, it will be important to determine whether airway tissue remodeling is specifically observed in the airway tissue of COVID-19 patients with poor outcomes (or in different animal models) in the early stages of infection. If such remodeling is confirmed, it may contribute to the elucidation of the crosstalk mechanism between immune cells and lung parenchymal cells that causes severe lung inflammation. In addition, the enhanced de novo synthesis pathway of nucleotides and the blood leakage of related molecules suggested in this study and other studies may be established as biomarkers with clear mechanisms of production. Elucidating the type of tissue remodeling that occurs in lung parenchymal cells in the early stages of respiratory disease and induces severe pneumonia in the later stages will be an important research target in future studies.

In summary, this study demonstrates that elevated serum concentrations of amino acid catabolites observed in the early stages of SARS-CoV-2 infection correlate with future severe disease onset. Experiments with respiratory virus-infected mouse models also show that the remodeling of pulmonary airway epithelial cells and vascular smooth muscle cells in the early stages of infection is responsive to changes in serum metabolites. Our findings suggest the possibility that transient tissue remodeling in the early stages of infection, followed by its subsequent disappearance, could trigger a series of pathological events culminating in severe pneumonia weeks later. This hypothesis warrants further exploration in future investigations.

## Methods
### Participants
Residual serum samples for biochemical and immunological tests were collected from patients with suspected SARS-CoV-2 infection who underwent RT-PCR testing of nasopharyngeal swab or saliva samples at Keio University Hospital or Osaka Metropolitan University Hospital from March 2020 to January 2021. The diagnosis of COVID-19 was based on a positive RT-PCR test from a nasopharyngeal swab or saliva sample. The classification of severity was based on the Japanese Ministry of Health, Labour and Welfare's classification of severity (criteria for evaluation by healthcare professionals) and was defined as follows: asymptomatic, no symptoms associated with COVID-19; mild disease, SARS-CoV-2-positive with no findings of pneumonia; moderate disease, SARS-CoV-2-positive with findings of pneumonia but not requiring a ventilator or intensive care unit (ICU) management; and severe disease, SARS-CoV-2-positive requiring ventilator or ICU management, including death attributed to COVID-19 during the treatment period. For a negative control, residual serum samples were collected from patients who underwent RT-PCR testing using nasopharyngeal swab fluid between April and May 2020 at Keio University Hospital and received negative RT-PCR test results, making them clinically unlikely

to have COVID-19. The study included the early patient cohorts of waves 1, 2, and 3 of COVID-19 conducted in Japan from March to December 2020. For cohorts 1 and 2, patients were not excluded because of comorbidities, as the main objective was to include the greatest possible number of severe cases for which specimens were available early after disease onset. Cases in the mild and moderate groups were selected based on matching age and sex distribution with the severe group. No specific exclusions were made in any group during the study period, mainly because the number of samples was limited. Asymptomatic and mildly symptomatic patients were selected for inclusion by balancing the sex ratio. The higher proportion of males in the moderate and severe categories is due to the observed tendency for males to be more severely affected[50]. This was particularly true for patients hospitalized during the study period. For surface symptoms and reported blood parameters of patients in this study, please refer to Supplementary Table 1 and Supplementary Fig. 16.

Sample collection and utilization were conducted under the approval of the Ethics Committee of the Keio University School of Medicine (approval numbers 20200059 and 20200063) and the Ethics Committee of Osaka Metropolitan University Graduate School of Medicine (approval number 2020-003). The use of residual samples was conducted on an opt-out basis based on the approval of the relevant ethics committees. All samples were stored at −80 °C.

### Clinical study design
Comprehensive metabolome and cytokine analyses were performed using serum samples from COVID-19 patients collected from multiple institutions for the following two purposes (Fig. 1, Supplementary Table 1):

#### (Cohort-1) Identification of prognostic and predictive metabolic markers for use in the early stages of infection (within 5 days of onset)
Patients with severe pneumonia during stage III of COVID-19 may exhibit an early inflammatory response in the lungs that differs in both quality and extent from that in patients with mild or moderate disease. This is probably due to the increased exposure to the virus and an underlying inflammatory response due to pre-existing conditions. To screen for serum metabolites that reflect differences in initial viral infection response, we performed a metabolomic analysis of sera collected from patients within 5 days of disease onset. We aimed to use the obtained metabolomic profiles of over 300 compounds to examine whether the serum metabolite compositions of healthy, asymptomatic, mild, moderate, and severe patients differed, and to extract candidate prognostic markers that characterized the severe group. We also performed a cytokine panel analysis of the same samples to identify cytokines that correlated with metabolite dynamics and to estimate the inflammatory pathways of lung parenchymal cells and/or immune cells associated with changes in serum metabolites.

#### (Cohort-2) Time-course profiling of serum metabolome changes in patients with different outcomes during the clinical course of COVID-19
Each stage of COVID-19 is thought to trigger a sequential inflammatory response with different mechanisms, ranging from initial infection to remission or severe pneumonia. These inflammations and their resolution processes are thought to be different in asymptomatic, mild, moderate, and severe cases. Blood samples collected serially from COVID-19 patients were analyzed to determine the changes in serum metabolite profiles between healthy, asymptomatic, mild, moderate, and severe cases with the aim of understanding how metabolic changes in individuals with severe disease differ from those in others. Finally, the temporal variability of the outcome markers obtained in Cohort-1 was identified to confirm their validity as prognostic markers in the early stage of the disease.

## Sample preparation of serum for metabolome analysis

Frozen serum samples of Cohort-1 participants, together with internal standard compounds (see below), were sonicated in ice-cold methanol (500 μL), to which an equal volume of chloroform and 0.4 times the volume of ultrapure water (LC/MS grade, Wako Pure Chemical, Tokyo, Japan) were added. The suspension was centrifuged at $15,000 \times g$ for 15 min at 4 °C. The aqueous phase was then filtered in an ultrafiltration tube (Ultrafree MC-PLHCC, Human Metabolome Technologies, Tsuruoka City, Japan), and the filtrate was concentrated using a vacuum concentrator (SpeedVac; Thermo Fisher Scientific, Waltham, MA, USA). The concentrated filtrate was dissolved in 50 μL ultrapure water and subjected to IC-HR-MS analysis. Methionine sulfone (L-Met) and 2-morpholinoethanesulfonic acid (MES) were used as internal standards for cationic and anionic metabolites, respectively. The loss of endogenous metabolites during sample preparation was corrected by calculating the recovery rate (%) of the standards in each sample measurement.

## SARS-CoV-2 in vivo infection model

SARS-CoV-2 for mice (SARS-CoV2-MA10) was generated via circular polymerase extension reaction[30,51]. BALB/cCrSlc (Balb/c) or C57BL/6 NCrSlc (C57BL/6) female mice (8–12 weeks) were purchased from Japan SLC, Inc (Hamamatsu, Japan). All animal experiments were conducted with the approval of the Animal Research Committee of Research Institute for Microbial Disease in Osaka University and performed in animal BSL-3 facilities at Osaka University (approval number R02-01-1). Mice were anesthetized by ketamine/xylazine and intranasally infected with SARS-CoV2-MA10 ($3 \times 10^4$: 50% tissue culture infectious dose/mouse) for 3 days. Body weight was measured every 24 h. Mice were euthanized at 3 days after infection, and lung tissues were collected.

## Heat fixation of SARS-CoV-2-infected tissues

Lung tissues were collected from four SARS-CoV-2-MA10-infected mice. These tissues were promptly divided into two halves: one half was subjected to heat fixation and the other half immersed in paraformaldehyde for subsequent immunohistochemistry application. Heat fixation was performed using a T1 Stabilizer (Denator, Gothenburg, Sweden)[52]. The heat fixation process of this instrument can completely inactivate both viral and bacterial pathogens, and the fixed samples remain robust to techniques such as MALDI-IMS and histological staining[53]. Heat fixation also rapidly inactivates metabolic enzymes, allowing the retention of metabolites such as lysophosphatidic acid, which are susceptible to post-mortem enzymatic degradation[54]. In contrast, methods such as focused microwave irradiation can achieve higher retention efficiencies for metabolites such as ATP that are more susceptible to enzymatic degradation[55,56]. However, there is no precedent for using this approach for lung fixation, and given the large size of the equipment and the challenges of moving it to a BSL3 facility, a portable heat-denaturing fixation system was used. The temperature was set to 95 °C for 30 s (optimized for fresh-frozen tissues) and top heater position to 90%. Tissues were then placed on the stabilizer card (Denator) for heat fixation according to the instrument guidelines. This animal experiment was approved by the Animal Experimentation Committee of Osaka University (Approval No. R02-01-1).

## Influenza virus in vivo infection model

All mice were housed at 21–22 °C under a 12-h alternating light–dark cycle. Influenza virus strain A/Puerto Rico/8/34 (A/PR8) was kindly provided by Dr. Takeshi Ichinohe (University of Tokyo). Nine-weeks-old C57BL/6 J female mice were fully anesthetized and then infected intranasally with 250 PFU of A/PR8 in a total volume of 30 μL. For anesthesia, a mixture of medetomidine (0.3 mg/kg; Nippon Zenyaku Kogyo Co., Ltd, Koriyama, Japan), midazolam (4 mg/kg; Astellas,

Tokyo, Japan), and butorphanol (5 mg/kg; Meiji Seika Pharma Co., Ltd., Tokyo, Japan) was intraperitoneally (i.p.) injected into the mice. The effects of medetomidine were reversed with the i.p. injection of atipamezole (0.3 mg/kg, Nippon Zenyaku Kogyo). The mice were sacrificed at 4 days after infection and tissue samples were collected. This animal experiment was approved by the Animal Studies Committees of Keio University.

## Metabolite extraction for tissue-based metabolomics

Metabolite extraction from lung tissue for comprehensive metabolomic profiling was performed using the slightly modified protocol of a previously established method[56]. Briefly, frozen tissue samples were combined with a 500-μL aliquot of ice-cold methanol containing methionine sulfone (L-Met) and 2-morpholinoethanesulfonic acid (MES), serving as designated internal standards (IS) for cationic and anionic metabolites, respectively. This mixture was subjected to homogenization using a Finger Masher manual homogenizer (AM79330, Sarstedt, Tokyo, Japan). To this homogenate, half the volume of ultrapure water (LC/MS grade, obtained from Wako) and 0.4 times the initial volume of chloroform (Nacalai Tesque, Kyoto, Japan) were added. The resulting mixture was centrifuged at $15,000 \times g$ for 90 min at a temperature of 4 °C. After centrifugation, the resulting aqueous phase was subjected to filtration using an ultrafiltration tube (Ultrafree MC-PLHCC, Human Metabolome Technologies, Yamagata, Japan). The filtrate was then concentrated by nitrogen flow-assisted evaporation on a heating block (DTU-28N, TAITEC, Koshigaya City, Japan). The concentrated filtrate was resuspended in 50 μL ultrapure water for subsequent LC-MS/MS and IC-HR-MS analytical procedures.

## Ion chromatography-high resolution mass spectrometry (IC-HR-MS) metabolome analysis

Metabolites were detected using an orbitrap-type MS instrument (Q-Exactive focus; Thermo Fisher Scientific) connected to a high-performance IC system (ICS-5000 + , Thermo Fisher Scientific) that enabled highly selective and sensitive metabolite quantification owing to the IC separation and Fourier transfer MS principle[57]. The IC instrument was equipped with an anion electrolytic suppressor (Dionex AERS 500; Thermo Fisher Scientific) to convert the potassium hydroxide gradient into pure water before the sample entered the MS instrument. Separation was performed using a Dionex IonPac AS11-HC 4 μm particle size column (Thermo Fisher Scientific). The IC flow rate was 0.25 mL/min, supplemented post-column with 0.18 mL/min makeup flow of MeOH. The potassium hydroxide gradient conditions for IC separation were as follows: from 1 mM to 100 mM (0–40 min) to 100 mM (40–50 min) and to 1 mM (50.1–60 min) at a column temperature of 30 °C. The mass spectrometer was operated in the ESI-negative mode for all detections. A full mass scan (m/z 70–900) was performed at a resolution of 70,000. The automatic gain control target was set at $3 \times 10^6$ ions, and the maximum ion injection time was 100 ms. The source ionization parameters were optimized with a spray voltage of 3 kV, and other parameters were as follows: transfer temperature, 320 °C; S-Lens level, 50; heater temperature, 300 °C; sheath gas, 36; and aux gas, 10.

## Liquid chromatography-tandem mass spectrometry for amino acid measurement

Cationic metabolite concentrations were determined by liquid chromatography-tandem mass spectrometry (LC-MS/MS) following published protocols[58]. In essence, we employed a triple-quadrupole mass spectrometer equipped with an electrospray ionization (ESI) ion source (LCMS-8060, Shimadzu Corporation, Kyoto, Japan) operated in both positive and negative-ESI and in multiple reaction monitoring (MRM) modes. Analyte separation was achieved on a Discovery HS F5-3 column (2.1 mm I.D. × 150 mm L, 3 μm particle size; Sigma-Aldrich, St. Louis, MO, USA) through a gradient elution with mobile phase A (0.1% formate) and mobile phase B (acetonitrile containing 0.1% formate).

The elution profile was as follows: 100:0 (0–5 min), 75:25 (5–11 min), 65:35 (11–15 min), 5:95 (15–20 min), and 100:0 (20–25 min), with a constant flow rate of 0.25 mL/min and a column oven set at 40 °C. The specific MRM conditions for each amino acid are shown in Supplementary Data 4.

## Quantitative and qualitative data analyses of targeted metabolome analysis

For the measurement of metabolites registered in our in-house compound library, we compared the measurement results of samples and corresponding standards and confirmed that the retention times were consistent. Chromatographic peak integrations and confirmation of signal specificity for target compounds were performed for IC-HR-MS and LC-MS/MS, respectively, using Trace Finder software (ver. 4.1, Thermo Fisher Scientific) and Lab Solutions software (ver. 5.113, Shimadzu). For IC-HR-MS data analysis, the Trace Finder compound identification and confirmation setup parameters included a molecular ion intensity threshold override of 10,000, S/N 5, and mass tolerance of 5 ppm. Isotope pattern analysis using a 90% fit threshold, 30% allowable relative intensity deviation, and 5 ppm mass deviation were also performed to ensure that the relative intensities of the $M + 1$ and/or $M + 2$ isotope peaks for each compound were consistent with the theoretical relative intensities. For LC-MS/MS analysis, chromatographic peaks obtained with compound-specific SRM channels (Supplementary Data 4) were integrated and manually reviewed. For a single target compound, one or more confirmatory SRM channels were set (if available) to confirm peak compound identification. Chromatograms were acquired using Lab Solutions (ver. 5.113, Shimadzu). Peak areas were determined using Data browser software. The obtained peak quantitation values for each compound were corrected for recovery by IS (MES and L-Met for IC-HR-MS and LC-MS/MS, respectively). For the tissue metabolome, peak area values were further corrected for weight.

## Multivariate statistical analysis

Supervised partial least squares discriminant analysis (PLS-DA) and HCA were performed to validate differences in the serum metabolome between groups and to identify significantly contributing metabolites. Analyses were performed using Metaboanalyst (v4.0), a web-based multivariate analysis tool[59]. For normalization, samples were subjected to median adjustment to uniformly correct for systematic differences across samples. Autoscaling was applied to standardize comparisons of variables, without performing data transformation. The validity of PLS-DA was assessed by multiple correlation coefficient ($R^2$) and cross-validation $R^2$ ($Q^2$) in addition to 10-fold cross-validation (see Supplementary Fig. 17 for details).

## ROC curve analysis

ROC curve-based analysis was then performed to identify potential biomarkers and evaluate their performance. The relative intensities of the compounds were directly entered into the analysis without normalization or scaling. ROC analysis for individual compounds was performed using Metaboanalyst (v4.0). Multivariate ROC curves and AUCs shown in Fig. 5d were derived using the "glm" function in R (version 4.2.3).

## RNA preparation and RT-qPCR

RNA was extracted from lung tissues using NucleoSpin RNA Plus (TaKaRa Bio, Kusatsu, Japan) for influenza-infected mice and TRIzol Reagent for SARS-CoV-2-infected mice. cDNA was synthesized using the ReverTra Ace qPCR RT Master Mix kit with gDNA Remover (TOYOBO, Osaka, Japan) following the manufacturer's protocol. RT-qPCR was performed using the StepOnePlus system (Thermo Fisher Scientific) with THUNDERBIRD SYBR qPCR Mix (TOYOBO). The relative expression of mRNA was determined using the ΔΔCt method. The oligonucleotide primers are listed in Supplementary Table 5.

## Luminex assay

A Human Premixed Multi-Analyte Kit (catalog # LXSAHM-23; R&D Systems, Minneapolis, MN, USA) was used to analyze CCL17/TARC, CXCL10/IP-10/CRG-2, IFN-a, IFN-g, IL-1b/IL-1F2, IL-2, IL-5, IL-10, IL-18/IL-1F4, IL-28B/IFN-l3, IL-33, TNF-a, CXCL9/MIG, GM-CSF, IFN-b, IL-1a/IL-1F1, IL-1ra/IL-1F3, IL-4, IL-6, IL-13, IL-28A/IFN-l2, IL-31, and lymphotoxin-a/TNF-b concentration. The Luminex assay was performed according to the manufacturer's instructions. Briefly, serum samples were centrifuged at $16,000 \times g$ for 4 min before use. The samples were diluted two-fold with the dilution buffer included in the kit, and antibodies specific to the analyte on magnetic particles embedded with the fluorophore were precoated at a set ratio for each analyte. Next, 50 μL of microparticles, standards, and samples were added to the wells, followed by incubation for 2 h at room temperature. A biotin-labeled detection antibody cocktail was added to the kit (Cocktail A: PART#893899, Cocktail B: PART#893901, Cocktail C:PART#894368, Cocktail D: PART#893985, Cocktail E: PART#893986, Cocktail G: PART#894625, Cocktail N: PART#894908, Cocktail O: PART#898124, Cocktail R: PART#898127, and Cocktail 7: PART#898551) for 1 h at room temperature, and the analytes were sandwiched between the capture and detection antibodies. Then, 50 mL of streptavidin conjugated with phycoerythrin was added and bound to the biotinylated detection antibody for 30 min at room temperature. The magnet in the analyzer captured and held superparamagnetic microparticles in a monolayer. Two light-emitting diodes (LEDs) with different spectra were used to identify the region by exciting the dye in each particle, and a second LED excited the streptavidin-phycoerythrin conjugate (streptavidin-PE) to measure the amount of analyte bound to the particle. Fluorescence intensities were obtained using a Luminex 100/200 analyzer (Luminex, Austin, TX, USA).

## ELISA

This assay was performed using a quantitative sandwich enzyme immunoassay method (catalog #D28B00, R&D Systems). The analyses were performed according to the manufacturer's instructions. Monoclonal antibodies specific to human IL-28B/IFN-l3 were prefixed to the microplate. A Quantikine Human Immunoassay (R&D Systems) was used to detect IL-28B/IFN-l3 concentration in patient serum samples, and absorbance was measured at 450 nm using a microplate reader.

Next, 100 mL each of standard, control, and samples were added to a precoated well and incubated for 2 h at room temperature. After removing the unbound substances, 200 mL of an enzyme-linked monoclonal antibody specific for human IL-28B was added to the wells and incubated for another 2 h at 2–8 °C. After removing unbound antibodies and recombinant enzymes by washing, 200 mL of substrate solution was added to the wells and the resulting color change was observed according to the amount of IL-28B/IFN-l3 bound in the first step. Stop solution (50 mL) was added to the wells, and the optical density of each well was detected using a microplate reader set at 450 nm.

## Histological analysis

SARS-CoV-2-infected lung tissues were fixed in 4% (w/v) paraformaldehyde at 4 °C. Fixed tissues were incubated with 10% (w/v) sucrose (Nacalai Tesque) in phosphate-buffered saline overnight at 4 °C and embedded in OCT compound (Sakura Finetek, Tokyo, Japan). After freezing, cryostat sections were cut to 5 mm thickness at −12 °C. Frozen sections were stained with H&E. For immunostaining, sectioned samples were fixed in cold acetone for 10 min at 4 °C and permeabilized in 100% methanol for 10 min at room temperature. After nonspecific antigens were blocked with 5% (w/v) skimmed milk for 30 min at room temperature, tissues were incubated in blocking solution with primary antibodies overnight at 4 °C. Primary antibodies were Prosurfactant Protein C (catalog #H00006440-M01; Novus Biologicals, Centennial, CO, USA) and CXCL10 (catalog #BS-1502R; Bioss,

Woburn, MA, USA), each diluted 1:500 in 1x PBS. Secondary antibodies were two Alexa Fluor-conjugated antibodies, Alexa Fluor 647 (anti-Armenian hamster, IgG; Jackson Immuno Research, Ely, UK) and Alexa Fluor 488 (anti-Rabbit, IgG; Abcam, Cambridge, UK), labeled at 1:500 dilution for 1 h at room temperature. For nuclear staining, tissues were incubated in 1:100 diluted DAPI solution (Nacalai Tesque). Immuno-fluorescence images were obtained using a fluorescence microscope (BZ-X800, 20× magnification; KEYENCE, Osaka, Japan) and analyzed with BZ-X800 Analyzer software (KEYENCE).

### Sample preparation for IMS

For IMS, lung tissues from heat-fixed SARS-CoV-2-infected or fresh-frozen influenza-infected mice were embedded in super cryoembedding medium (SCEM; SECTION LAB, Hiroshima, Japan) and then stored at −80 °C until analysis. The tissue-containing SCEM blocks were sectioned at a cryostat temperature of −16 °C into 8-μm-thick sections using a CM 3050 cryostat (Leica, Wetzlar, Germany). These sections were transferred to indium tin oxide (ITO)-coated glass slides (Matsunami Glass Industries, Osaka, Japan) for further analyses. The mounted sections were manually coated with a matrix solution containing 9-aminoacridine (10 mg/mL dissolved in 80% ethanol) using an art brush (Procon Boy FWA Platinum, Mr. Hobby, Tokyo, Japan). The matrix solution was applied from a distance of approximately 15 cm, with approximately 1 mL sprayed on each slide. To maintain uniform conditions for analyte extraction and co-crystallization, matrix application was performed simultaneously on multiple slides, as previously described[60]. Optical images of lung sections were obtained by a scanner and subjected to MALDI-MS imaging.

### IMS

Matrix-assisted laser desorption/ionization (MALDI) imaging was performed using a Bruker timsTOF fleX MS (Bruker Daltonics, Bremen, Germany) operated in quadrupole time-of-flight (qTOF) analysis mode. The acquisition parameters were set as follows: negative ion mode detection, pixel resolution of 80 μm, 200 laser pulses per pixel at a frequency of 10 kHz, and laser power setting of 50%. Data acquisition was targeted to an m/z range of 100–650. The raw mass spectra were processed and reconstructed using SCiLS Lab (v. 2019, Bruker Daltonics), which allowed the generation of detailed MS images. Signals within the targeted range were normalized to the total ion current to account for variations in ionization efficiency among pixels. The imaged metabolites were primarily identified as nucleotides and related molecular species, following previously published protocols[55]. Metabolite identifications were validated based on accurate mass measurements, and congruence checks with reference standards analyzed by MALDI-MS were used to further confirm the m/z consistency of the identified ions.

### Statistical analysis

Statistical analyses were performed using Prism v. 9.5.1 (GraphPad Software, Inc., San Diego, CA, USA). The analyses included unpaired Student's t-test, one-way ANOVA with Tukey's multiple comparisons test, and/or two-way ANOVA with Tukey's multiple comparisons tests, as specified in each figure legend.

### Reporting summary

Further information on research design is available in the Nature Portfolio Reporting Summary linked to this article.

## Data availability

The metabolomics raw data from COVID-19 patients or healthy control serum have been deposited in Metabolomics Workbench under the DOI for this project (ST002984): https://doi.org/10.21228/M8DT65. All data generated or analyzed in this study are available with this article and its supplemental files. Supplementary Data 1 is the quantitative data for water-soluble metabolites obtained by IC-HR-MS and LC-MS/MS. Supplementary Data 2 is the quantitative data for lipidome analysis obtained by LC-HR-MS. Supplementary Data 3 shows the quantitative data for steroids obtained by LC-MS/MS. The MRM library of compounds measured by LC-MS/MS is available as Supplementary Data 4. Other data sets used and/or analyzed during the current study are available from the corresponding author upon reasonable request. Source data 1 shows serum cytokine quantification data from cytokine panel analysis. Source data are provided with this paper.

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

## Acknowledgements

This research was supported by the Japan Society for the Promotion of Science KAKENHI (22K15927 to Y.N.), the Japan Agency for Medical Research and Development (JP20jk0110021 to Y.N., and JP20he1122001 and JP20wm0125003 to Y.Kido), the Osaka Metropolitan University Strategic Research Grant (OCU-SRG2021_YR09 to Y.N.), the COVID-19 Private Fund (to Y.Kido), Moonshot Research & Development (JPMJMS2025 to M. Sasai, M.Y., Y.M., 22zf0127007s0301 to M.S.) and All-Osaka U Research in "The Nippon Foundation - Osaka University Project

for Infectious Disease Prevention," AMED-SCARDA project for Vaccine Development (Toshio Ito as the Lead) as a collaborator in Central Institute for Experimental Animals (M. Suematsu), the Japan Agency for Medical Research and Development (23zf0127007s0102, JP23zf0127003, 22fk0108511h0001 and JP23gm1210009 to Y.S.), JSPS KAKENHI (22H02833 to Y.S.).

## Author contributions

R.M., N.T., and Y.S. performed the metabolome and histological analysis. N.K.S., M.M., M.Su., M.Yan., and Y.S. wrote the manuscript. Y.U. M.W. Y.N. and Y.Kido collected samples for a clinical study. M.Sa., M.Yam., Y.M. J.U., G.Y., M.H., S.T., and Y.Kim performed the infection experiment. Y.S. conceptualized the study and provided methodology and investigation.

## Competing interests

The authors declare no competing interests.
