## [Peer Review File · Nature Communications]

Amino acid catabolite markers for early prognostication of pneumonia in patients with COVID-19Reviewers' Comments:

Reviewer #1:

Remarks to the Author:

In this manuscript, Maeda et al identify amino acid catabolism as an early biomarker of COVID-19 disease severity, and the biological mechanisms involved. There is a strong need for such biomarkers. Overall, this is a well-designed and well-written manuscript. The mouse model data are particularly important to deconvolute the human findings from any possible clinical confounders such as comorbidities. However, I have the following concerns:

Major issues:

1. Details of the heat treatment prior to imaging mass spectrometry should be provided (temperature, duration, validation with regards to metabolite stability and to successful viral inactivation/biosafety requirements).
2. Details are lacking for MALDI analysis (matrix used and parameters for matrix application, laser power, mass spectrometer detection parameters).
3. Metabolite extraction methods for lung tissue are lacking.
4. Metabolomics data processing parameters are lacking (software used, version, noise, deconvolution, alignment, normalization method, etc).
5. Approach used for metabolite annotation and its confidence are lacking.
6. Statistical analysis details are incomplete (for example, the paragraph in Methods does not mention the approaches used for hierarchical clustering or PLSDA). Methods for ROC curve generation, especially combinatorial ROC analysis, are also lacking.
7. I strongly recommend that authors make their metabolomics data publicly available.
8. Reporting summary page 3: in subsection Materials and Experimental Systems, clinical data should be checked and necessary information provided.
9. Manuscript states that patient sera was collected within 5 days. Given the rapid progression of infection and disease (and expected concomitant metabolic changes), if available, data should be provided on the specific day post-disease-onset that samples were collected. Table 1 may be a good place to add this information.

Minor issues:

1. PLSDA cross-validation is necessary. Alternatively, authors may choose instead to use a method less prone to over-fitting, such as PCA or PCoA.
2. Page 5: "By contrast, levels of serum serotonin, which likely is released from the platelets, decreased (Supplementary Fig. 3a, b)." This statement is slightly misleading, since no significant difference in serotonin is observed between negative or asymptomatic patients vs severe patients.
3. Supplementary figure 3b: significance stars appear to have been cut off.
4. Histidine and proline data in supplementary Figure 3 do not show strong differences between groups. I suggest that discussion of these either be tempered or removed.
5. Figure 4a: it would be helpful to mark the metabolites listed onto the hierarchical clustering map, so that readers can assess the variability.
6. Page 8 typo: "Although blood levels of the Leu did not change" should read "Although blood levels of Leucine did not change"
7. Page 8: "for example, levels of creatine and creatinine, which are thought to originate from muscle tissue 25, were elevated in the late stage of the disease (Fig. 4o)". Data is only provided for creatine. Figure should be amended to provide creatinine data, or text should be modified.
8. Page 22: "Fifty milliliters of microparticles, standards, and samples were added to the wells, followed by incubation for 2 h at room temperature." Should this read "Fifty microliters"?
9. Page 25: "Optical images of brain sections were obtained" should read: "Optical images of lung sections"
10. Page 6: "the levels of glutamine (Gln), the amino acid with the highest blood concentration". A reference would be helpful to support this statement.
11. Fig. 3a-f, please define a.u. in figure legend
12. Table 3: please clarify the meaning of the bolding in Table 3.

13. Figure 4: graphs showing the averages per-group would be helpful for Thr, Leu, Kyn, perhaps in Figure S4

Reviewer #2:

Remarks to the Author:

Reviewer #3:

Remarks to the Author:

The manuscript „Amino acid catabolite markers for early prognostication of pneumonia in patients with COVID-19“ by Maeda and co-workers examines an important question, namely early predictive markers and underlying biochemical pathomechanisms for a severe course of COVID-19.

By serum metabolome and cytokine analyses of COVID-19 patients (collected from multiple centers in Japan) changes of amino acid metabolism were identified as prognostic markers. Concentrations of several amino acids were correlated strongly with a high viral load, serum concentrations of the tryptophan catabolite kynurenine and adenyly succinate were suited best to differentiate between different courses of disease. Combined cytokine and metabolite analyses indicated a strong activation of the IFN γ -CXCL10/9 axis, which was then further examined by animal experiments.

In these animal experiments disease severity, lung metabolomic remodeling and cytokine production were investigated in mice infected with SARS-CoV2-MA10 and influenza virus. mRNA expression of Ifng, Cxcl10, Cxcl9, IDO-1, IDO-2, Bcat-1 and Ki67 was significantly higher in the severely affected Balb/c mice (both in mice infected with SARS-CoV2-MA10 and influenza virus) compared to the mild disease model- suggesting that the inflammatory response to SARS-CoV2-MA10 infection is similar to that in influenza infection.

Lung metabolomic analyses confirmed enhanced amino acid catabolism (along the kynurenine pathway and BCAA deamination) and de novo nucleotide synthesis pathway activity in the severely affected Balb/c mice compared to mildly affected B6 mice.

Abnormal proliferation of mucosal secretory cells and smooth muscle cells was found in both SARS-CoV2-MA10 and influenza mouse models, indicating similar remodeling of the lung in response to early respiratory virus infections.

The manuscript is of significance for the field and shows very interesting, novel data and should be published after minor modifications.

The study design and the used methods are appropriate, the manuscript is written very well and the conclusions drawn are also supported by the data.

These are my questions/comments:

- The number of patient samples with mild, moderate and severe course of disease of the different centres should be mentioned clearly in the main manuscript/Abstract. Actually, cohort-2 only includes 12 individuals, in which time course analyses were performed. Did time course analyses show similar patterns in men and women?

(There appear to be gender-related differences regarding amino acid concentrations, which might be aggravated during severe disease)

- Please also state clearly, what the inclusion/exclusion criteria were: from the table it appears, that patients had maximally two/three co-morbidities. Actually, the number and also the extent of co-

morbidities may of course also influence the course of disease- patients with more co-morbidities probably have more "background" inflammation/ higher stress (cortisol baseline) levels and secondary immunodeficiency might be a problem.

You might e.g. calculate, whether there were differences regarding the investigated parameters in patients without/with co-morbidities- especially in patients with a more severe course of disease.

- According to the supplementary data presented also disturbances of hormones/neurotransmitters appear to be a major problem in Covid-19 infected patients with a more severe outcome: probably patients with a severe course of Covid-19 need higher cortisol levels to fight the infection/compensate, resulting in a reduced formation of other steroid hormones. Secondary adrenal insufficiency might in fact compromise the host response to infection significantly and might be the reason, why patients develop pneumonia/have a severe course of disease- Did you also measure levels of steroid hormones in the mild/severe disease model? And did you also determine catecholamine levels and insulin levels in patients (Phenylalanine and tyrosine were correlated negatively with viral load, many patients developed prediabetes or diabetes and insulin is an antagonist of cortisol)?

- The correlations between homocysteic acid, ADMA and viral load could indicate endothelial dysfunction. Did you also investigate other markers of endothelial dysfunction/microclot formation or oxidative/nitrosative stress? Oxidative stress is also induced by IFN-g, and depletion of antioxidants (like glutathione, vitamin C and vitamin E; but also of B vitamins- and vitamin B6 was low) can go along with impaired cell function and also impaired neurotransmitter formation. If you have data, please add or otherwise discuss.

- Were there any differences regarding the investigated parameters between patients receiving oxygen treatment or intubation and those who did not? And did you record other symptoms patients had during acute Covid-19 (e.g. neurological symptoms, chest pain, dyspnea, strong fatigue, sleep disturbances, depression/anxiety, pain,...)? If yes, were there any differences regarding amino acids in patients with/without symptoms?

(Enhanced tryptophan catabolism can impair serotonin and melatonin formation and serotonin levels were lower in patients with more severe disease, this might explain many symptoms that hospitalized patients, but also Post Covid patients have, e.g. sleep disturbance, depressed mood, anxiety, fatigue).

- Please also provide data of hemoglobin, iron parameters, serum interleukin-6, ferritin and ideally neopterin concentrations of patients (if available) and in case discuss them. (Earlier studies suggested that these parameters were also predictive for a worse outcome of patients: e.g.

<https://doi.org/10.3390/metabo11100653>; <https://doi.org/10.1021/acs.jproteome.1c00052>;

<https://doi.org/10.1186/s12879-020-05671-7>; <https://doi.org/10.1093/ofid/ofaa521>;

<https://doi.org/10.1016/j.jphs.2022.06.005>;

-I would also suggest to discuss very recent papers investigating metabolomic approaches in patients with acute Covid-19 and the role of IDO in orchestrating immune response.

Reviewer #4:

Remarks to the Author:

The experimental studies in mice are nicely done and appropriate controls are included. The conclusions relating to remodeling of lung tissue and hyperproliferation of certain cells in lungs are very convincing. While the comparison between high- and low-dose infection is legitimate it is unclear how any of this impacts the virus dissemination in lung tissue. The reader would benefit from being able to look at the virus load data in form of "infectious virus tissue titration" and not only immunohistology. At this stage it is unclear if there is potentially any antiviral effect not predicted in some of the groups. In other words, is there a difference in virus growth that contributes to pathological differences in these lungs? Otherwise, this is a very solid study that increases our understanding of corona virus pathology.

We thank the Reviewers for their comprehensive and constructive feedback on our manuscript. The recommendations and comments have improved the quality of the manuscript, and we have addressed all concerns raised in the review.

The following major revisions were made:

- (1) In response to reviewer 1's comments, we have expanded the Methods section and published the metabolomic dataset (including newly added Source data 1-4) to provide a more comprehensive description.
- (2) In response to Reviewer 2's comments, we have added new patient clinical data and included correlation analyses of serum metabolites and these data.
- (3) In response to Reviewer 3, we provided information on viral concentrations in animal infection studies.

Below, we provide specific responses to each of the Reviewers' concerns. Revised text is shown in blue font in the revised manuscript.

Reviewer #1 (Remarks to the Author):

In this manuscript, Maeda et al identify amino acid catabolism as an early biomarker of COVID-19 disease severity, and the biological mechanisms involved. There is a strong need for such biomarkers. Overall, this is a well-designed and well-written manuscript. The mouse model data are particularly important to deconvolute the human findings from any possible clinical confounders such as comorbidities. However, I have the following concerns:

Reply:

We thank the Reviewer for their support and insightful comments. We have responded to each of their specific comments in detail below. Based on the Reviewer's suggestion that essential information was missing from our methodology description, we have made substantial additions to the revised manuscript to address these concerns. We believe that these revisions have made our manuscript significantly more robust.

1. Details of the heat treatment prior to imaging mass spectrometry should be provided (temperature, duration, validation with regards to metabolite stability and to successful viral inactivation/biosafety requirements).

Reply:

We appreciate the Reviewer correctly pointing out that this information was missing. As noted, the description of the processing of SARS-CoV-2-MA10 infected animal tissues used for imaging MS and metabolomic analysis was not sufficiently detailed. This information is critical when analyzing metabolites from samples derived from infectious animals handled in BSL3.

In this study, we applied a heat treatment using a T1 heat stabilizer (Denator) ¹immediately after the animals were sacrificed in the P3 experimental facility. After the virus was inactivated, the tissues underwent metabolomic analysis and imaging mass spectrometry. Previous studies have demonstrated that heat fixation using this device can effectively deactivate viral and bacterial pathogens. Additionally, heat-fixed samples have proven to be resistant to procedures like MALDI-IMS and histological staining (e.g., HE staining)². Our

experiments using this device were therefore approved for use within the BSL3 facility of Osaka University (approval number R02-01-1).

In addition, we have included information on the stability of metabolites associated with this heat treatment method. Key points include the ability of heat fixation to quickly deactivate metabolic enzymes, thus preserving vulnerable metabolites like lysophosphatidic acid from post-mortem enzymatic degradation³. On the other hand, for metabolites such as ATP that are more susceptible to enzymatic degradation, methods such as focused microwave irradiation can achieve higher retention efficiencies⁴. However, this type of instrument is large, requires a greater electric power supply, and is difficult to move to a BSL3 facility, and we therefore chose to use the portable heat-denaturing fixation system in this study.

In the revised manuscript, the following detailed conditions and information have been added to the Methods section.

Heat fixation of SARS-CoV-2 infected tissues

Lung tissues were collected from four SARS-CoV-2-MA10-infected mice. These tissues were promptly divided into two halves: one half subjected to heat fixation and the other half immersed in paraformaldehyde for subsequent immunohistochemistry application. Heat fixation was performed using a T1 Stabilizer (Denator, Gothenburg, Sweden)¹. The heat fixation process of this instrument can completely inactivate both viral and bacterial pathogens, and the fixed samples remain robust to techniques such as MALDI-IMS and histological staining². Heat fixation also rapidly inactivates metabolic enzymes, allowing the retention of metabolites such as lysophosphatidic acid, which are susceptible to post-mortem enzymatic degradation³. In contrast, methods such as focused microwave irradiation can achieve higher retention efficiencies for metabolites such as ATP that are more susceptible to enzymatic degradation⁴. However, there is no precedent for using this approach for lung fixation, and given the large size of the equipment and the challenges of moving it to a BSL3 facility, a portable heat-denaturing fixation system was used. Temperature was set to 95 °C for 30 s (optimized for fresh-frozen tissues) and top heater position to 90%. Tissues were then placed on the stabilizer card (Denator) for heat fixation according to the instrument guidelines. This animal experiment was approved by the Animal Experimentation Committee of Osaka University (Approval No. R02-01-1).

2. Details are lacking for MALDI analysis (matrix used and parameters for matrix application, laser power, mass spectrometer detection parameters).

Reply:

We appreciate the Reviewer's constructive feedback. As pointed out, the Methods section for imaging MS lacked information on matrix application and settings of the imaging mass spectrometry instrument, rendering it incomplete. In the revised manuscript, the Methods section has been updated with the following additions.

Sample preparation for IMS

For IMS, lung tissues from heat-fixed SARS-CoV-2-infected or fresh-frozen influenza-infected mice were embedded in super cryoembedding medium (SCEM; SECTION LAB, Hiroshima, Japan) and then stored at -80 °C until analysis. The tissue-containing SCEM blocks were sectioned at a cryostat temperature of -16°C into 8-µm-thick sections using a CM 3050 cryostat (Leica, Wetzlar, Germany). These sections were transferred to indium tin oxide

(ITO)-coated glass slides (Matsunami Glass Industries, Osaka, Japan) for further analyses. The mounted sections were manually coated with a matrix solution containing 9-aminoacridine (10 mg/mL dissolved in 80% ethanol) using an art brush (Procon Boy FWA Platinum, Mr. Hobby, Tokyo, Japan). The matrix solution was applied from a distance of approximately 15 cm, with approximately 1 mL sprayed on each slide. To maintain uniform conditions for analyte extraction and co-crystallization, matrix application was performed simultaneously on multiple slides, as previously described⁵. Optical images of lung sections were obtained by a scanner and subjected to MALDI-MS imaging.

IMS

Matrix-assisted laser desorption/ionization (MALDI) imaging was performed using a Bruker timsTOF fleX MS (Bruker Daltonics, Bremen, Germany) operated in quadrupole time-of-flight (qTOF) analysis mode. The acquisition parameters were set as follows: negative ion mode detection, pixel resolution of 80 μm , 200 laser pulses per pixel at a frequency of 10 kHz, and laser power setting of 50%. Data acquisition was targeted to an m/z range of 100–650. The raw mass spectra were processed and reconstructed using SCiLS Lab (v. 2019, Bruker Daltonics), which allowed the generation of detailed MS images. Signals within the targeted range were normalized to the total ion current to account for variations in ionization efficiency among pixels. The imaged metabolites were primarily identified as nucleotides and related molecular species, following previously published protocols⁴. Metabolite identifications were validated based on accurate mass measurements, and congruence checks with reference standards analyzed by MALDI-MS were used to further confirm the m/z consistency of the identified ions.

3. Metabolite extraction methods for lung tissue are lacking.

Reply:

Again, we appreciate the Reviewer pointing out that this information is missing. In the revised manuscript, we have added the following details.

Metabolite extraction for tissue-based metabolomics

Metabolite extraction from lung tissue for comprehensive metabolomic profiling was performed using the slightly modified protocol of a previously established method⁶. Briefly, frozen tissue samples were combined with a 500- μL aliquot of ice-cold methanol containing methionine sulfone (L-Met) and 2-morpholinoethanesulfonic acid (MES), serving as designated internal standards (IS) for cationic and anionic metabolites, respectively. This mixture was subjected to homogenization using a Finger Masher manual homogenizer (AM79330, Sarstedt, Tokyo, Japan). To this homogenate, half the volume of ultrapure water (LC/MS grade, obtained from Wako) and 0.4 times the initial volume of chloroform (Nacalai Tesque, Kyoto, Japan) were added. The resulting mixture was centrifuged at $15,000 \times g$ for 90 min at a temperature of 4 °C. After centrifugation, the resulting aqueous phase was subjected to filtration using an ultrafiltration tube (Ultrafree MC-PLHCC, Human Metabolome Technologies, Yamagata, Japan). The filtrate was then concentrated by nitrogen flow-assisted evaporation on a heating block (DTU-28N, TAITEC, Koshigaya City, Japan). The concentrated filtrate was resuspended in 50 μL ultrapure water for subsequent LC-MS/MS and IC-HR-MS analytical procedures.

4. Metabolomics data processing parameters are lacking (software used, version, noise, deconvolution, alignment, normalization method, etc).

Reply:

We would like to thank the Reviewer for checking every detail and their constructive suggestions. In the revised manuscript, we have added missing information on the methodology of metabolomics for the information of the readers, and we have also published all data obtained in this study. In fact, an important part of the metabolomics methodology was not described in the original manuscript due to an error, and we are grateful to the Reviewer for this comment, which gave us the opportunity to correct this issue.

This study used complementary measurements of the same sample by IC-HR-MS and LC-MS/MS [pentafluorophenylpropyl (PFPP) column separation] for both blood and tissue metabolome analysis. Specifically, as in previous studies, IC-HR-MS was used to measure mainly anionic metabolites, focusing on organic acids, sugar phosphates, nucleotides, and some amino acids and nucleobases. LC-MS/MS with PFPP columns focused on cationic metabolites, complementing the measurement of amino acids and nucleobases to achieve comprehensive metabolome measurements⁷.

We have added the following details in the revised manuscript.

Liquid chromatography-tandem mass spectrometry for amino acid measurement

Cationic metabolite concentrations were determined by liquid chromatography-tandem mass spectrometry (LC-MS/MS) following published protocols⁷. In essence, we employed a triple-quadrupole mass spectrometer equipped with an electrospray ionization (ESI) ion source (LCMS-8060, Shimadzu Corporation, Kyoto, Japan) operated in both positive and negative-ESI and in multiple reaction monitoring (MRM) modes. Analyte separation was achieved on a Discovery HS F5-3 column (2.1 mm I.D. × 150 mm L, 3 μm particle size; Sigma-Aldrich, St. Louis, MO, USA) through a gradient elution with mobile phase A (0.1% formate) and mobile phase B (0.1% acetonitrile). The elution profile was as follows: 100:0 (0–5 min), 75:25 (5–11 min), 65:35 (11–15 min), 5:95 (15–20 min), and 100:0 (20–25 min), with a constant flow rate of 0.25 mL/min and a column oven set at 40 °C. The specific MRM conditions for each amino acid were as shown in Source data 4.

Source data 4: MRM channels for targeted metabolomics analysis by LC-MS/MS

Analyte	Retention time (min)	Polarity	MRM transition (1)		MRM transition (2)	
			Precursor ion	Product ion	Precursor ion	Product ion
2-Aminobutyric acid	3.0	(+)	104.1	41.1	104.1	58.1
3-Aminobutyric acid	3.1	(+)	104.2	86.0	104.2	45.0
3-Hydroxykynurenine	7.6	(+)	225.1	208.0	225.1	162.2
3-Hydroxyanthranilic acid	8.1	(+)	154.1	108.2	154.1	80.2
3-Indoleacetic acid	10.0	(+)	176.1	130.1	176.1	77.2
3-Indoleethanol	10.9	(+)	162.1	117.3	162.1	115.2
3,4-Dihydroxyphenylacetic acid	8.1	(-)	167.2	123.1		
4-Aminobutyric acid (GABA)	4.2	(+)	104.1	87.1	104.1	45.1
4-Hydroxyproline	2.3	(+)	132.1	68.1	132.1	86.1
5-Glutamylcysteine	3.4	(+)	251.1	122.1	251.1	84.1
5-Hydroxy-L-tryptophan	8.3	(+)	221.1	204.1	221.1	162.3
5-hydroxy-tryptophol	8.0	(+)	178.1	160.3	178.1	115.3
5-Hydroxyindole-3-acetic acid	8.2	(+)	192.1	146.2	192.1	91.0
5-Methyltetrahydrofolic acid	9.3	(+)	460.2	313.2	460.2	180.1
Acetylcarnitine	10.1	(+)	204.1	85.1	204.1	60.1
Acetylcholine	10.6	(+)	146.1	87.1		
Adenine	6.3	(+)	136.1	119.0	136.1	92.1
Adenosine	7.5	(+)	268.1	119.0	268.1	136.1
Adenylsuccinic acid	6.8	(+)	464.1	252.1	464.1	162.0
Agmatine	2.1	(+)	131.4	72.2	131.4	60.1
Alanine	2.6	(+)	90.1	45.0		
Anthranilate	9.8	(+)	137.6	120.3	137.6	92.2
Arginine	3.5	(+)	175.1	60.1	175.1	70.1
Argininosuccinic acid	3.4	(+)	291.0	70.1	291.0	116.1
Ascorbic acid	1.2	(-)	175.0	87.0	175.0	115.0
Asparagine	2.2	(+)	133.1	87.2		
Aspartic acid	2.2	(+)	134.0	74.1	134.0	88.1
Asymmetric dimethylarginine	7.8	(+)	203.2	46.1	203.2	70.2
b-Nicotinamide mononucleotide	2.8	(+)	335.1	123.1	335.1	97.2
beta-Alanine	1.4	(+)	90.2	72.2		
Biotin (Vitamine B7)	8.0	(+)	245.1	227.0	245.1	97.0
Carnitine	6.0	(+)	162.1	60.1	162.1	103.1
Carnosine	5.3	(+)	227.1	110.1	227.1	156.1
Choline	5.1	(+)	104.1	45.1	104.1	60.1
Citicoline	2.5	(+)	489.1	184.1	489.1	264.1
Citrulline	2.5	(+)	176.1	159.1	176.1	70.1
Creatine	3.9	(+)	132.1	90.1	132.1	44.1
Creatinine	5.4	(+)	114.1	44.2		
Cystathionine	2.3	(+)	223.0	88.1	223.0	134.1
Cysteine	2.4	(+)	122.1	59.0	122.1	76.1
Cystine	2.1	(+)	241.0	73.9	241.0	152.0
Cytidine	7.1	(+)	244.1	112.1	244.1	95.0
Cytosine	4.4	(+)	112.1	52.0	112.1	67.1
dihydrofolate	6.6	(+)	444.0	178.1	444.0	297.1
Dimethylglycine	2.5	(+)	104.1	44.1	104.1	58.1
Dopa	7.1	(+)	198.1	152.2		
Dopamine	9.2	(+)	154.1	137.1	154.1	91.1
Epinephrine	8.0	(+)	184.1	166.1	184.1	107.1
FAD	6.8	(+)	786.2	348.0	786.2	136.1
Folinic acid	7.3	(+)	474.1	327.1	474.1	299.1
Glutamic acid	2.6	(+)	148.1	102.1	148.1	56.1
Glutamine	2.4	(+)	147.1	84.2	147.1	130.1
Glutathione reduced form	5.3	(+)	308.1	76.1	308.1	179.1
Glutathione oxidized form	6.2	(+)	613.2	355.0	613.2	231.1
Glycine	2.3	(+)	75.9	30.2	75.9	31.1
Guanosine	6.9	(+)	284.0	152.0	284.0	135.0
Hippuric Acid	7.4	(-)	178.0	134.1	178.0	77.0
Histamine	5.5	(+)	112.1	95.1	112.1	41.2
Histidine	3.1	(+)	155.9	110.1	155.9	56.1
Homo-Serine	2.4	(+)	120.1	74.0		
Homocysteine	3.6	(+)	136.0	56.1	136.0	90.1
Homocystine	4.7	(+)	269.1	88.1	269.1	136.1
Homogentisic Acid	8.1	(-)	167.0	123.1	167.0	122.1
Hypoxanthine	4.5	(+)	137.0	110.0	137.0	55.1
Indole lactic Acid	9.3	(+)	206.1	188.1	206.1	160.1
Indole-3-butyric acid	14.8	(+)	204.2	186.2	204.2	130.3
Indole-3-propionic acid	11.8	(+)	190.3	130.2	190.3	77.2
Indole-acetamide	10.8	(+)	175.1	130.1	175.1	77.1
Indoxyl sulfate	7.3	(-)	212.0	80.0	212.0	81.0
Inosine	6.0	(+)	269.1	137.1	269.1	119.0
Isoleucine	8.1	(+)	132.1	86.1	132.1	30.1

Second, as the Reviewer correctly pointed out, there was an omission in the description of the data processing method used for quantitative metabolomics data analysis. We have consequently added the required information to the section describing the metabolomic analysis method.

Quantitative and qualitative data analysis of targeted metabolome analysis

For the measurement of metabolites registered in our in-house compound library, we compared the measurement results of samples and corresponding standards and confirmed that the retention times were consistent. Chromatographic peak integrations and confirmation of signal specificity for target compounds were performed for IC-HR-MS and LC-MS/MS, respectively, using Trace Finder software (ver. 4.1, Thermo Fisher Scientific) and Lab Solutions software (ver. 5.113, Shimadzu). For IC-HR-MS data analysis, the Trace Finder compound identification and confirmation setup parameters included a molecular ion intensity threshold override of 10,000, S/N(signal-to-noise ratio) 5, and mass tolerance of 5 ppm. Isotope pattern analysis using a 90% fit threshold, 30% allowable relative intensity deviation, and 5 ppm mass deviation was also performed to ensure that the relative intensities of the M + 1 and/or M + 2 isotope peaks for each compound were consistent with the theoretical relative intensities. For LC-MS/MS analysis, chromatographic peaks obtained with compound-specific SRM channels (Source data 4) were integrated and manually reviewed. For a single target compound, one or more confirmatory SRM channels were set (if available) to confirm peak compound identification. Chromatograms were acquired using Lab Solutions. Peak areas were determined using Data browser software (ver. 5.113, Shimadzu). The obtained peak quantitation values for each compound were corrected for recovery by IS(MES and L-Met for IC-HR-MS and LC-MS/MS, respectively). For the tissue metabolome, peak area values were further corrected for weight.

5. Approach used for metabolite annotation and its confidence are lacking.

Reply:

We agree with the Reviewer's comment that there is insufficient information on metabolite annotation methods. In the updated manuscript, we have added this description in the Methods section (Page 26, **Quantitative and qualitative data analysis of targeted metabolome analysis**), as indicated in the response to comment 4.

6. Statistical analysis details are incomplete (for example, the paragraph in Methods does not mention the approaches used for hierarchical clustering or PLS-DA). Methods for ROC curve generation, especially combinatorial ROC analysis, are also lacking.

Reply:

We acknowledge the oversight in omitting the description of multivariate and ROC analyses. The updated manuscript now includes the following additions as a Methods section.

Multivariate statistical analysis

Supervised partial least squares discriminant analysis (PLS-DA) and HCA were performed to validate differences in the serum metabolome between groups and to identify significantly contributing metabolites. Analyses were performed using Metaboanalyst (v4.0), a web-based multivariate analysis tool⁸ or normalization, samples were subjected to median adjustment to uniformly correct for systematic differences across samples. Autoscaling was applied to standardize comparisons of variables, without performing data transformation. The validity of

PLS-DA was assessed by multiple correlation coefficient (R^2) and cross-validation R^2 (Q^2) in addition to 10-fold cross-validation (see Supplementary Fig. 17 for details).

ROC curve analysis

ROC curve-based analysis was then performed to identify potential biomarkers and evaluate their performance. The relative intensities of the compounds were directly entered into the analysis without normalization or scaling. ROC analysis for individual compounds was performed using Metaboanalyst (v4.0). Multivariate ROC curves and AUCs shown in Fig. 5d were derived using the “glm” function in R (version 4.2.3).

7. I strongly recommend that authors make their metabolomics data publicly available.

Reply:

The Reviewer's point is valid, and the results of the following metabolomic measurements carried out in this study have been newly added as Supplementary information (see also Data Availability section).

1. Results of metabolomic analysis of hydrophilic compounds (Source data 1)
2. Results of lipidomic analysis of hydrophobic compounds (Source data 2)
3. Results of comprehensive steroid analysis (Source data 3)

In addition, technical information on the following measurements has been provided for easy reference by readers.

4. MRM library of compounds measured by LC-MS/MS (Source data 4)

Readers can find this information in the newly added Data Availability section.

8. Reporting summary page 3: in subsection Materials and Experimental Systems, clinical data should be checked and necessary information provided.

Reply:

This study is not a clinical trial, but a clinical observational study. Clinical Data in the Reporting Summary is the data reporting format used when conducting research on the subject of a clinical trial and does not apply to this study. Please refer to <https://www.nature.com/nature-portfolio/editorial-policies/ethics-and-biosecurity#bioethics-policy>, and <https://www.icmje.org/recommendations/browse/publishing-and-editorial-issues/clinical-trial-registration.html> .

9. Manuscript states that patient sera was collected within 5 days. Given the rapid progression of infection and disease (and expected concomitant metabolic changes), if available, data should be provided on the specific day post-disease-onset that samples were collected. Table 1 may be a good place to add this information.

Reply:

The Reviewer's point is reasonable. In Cohort 1, specimens were collected and incorporated observing the criterion that patients were within 5 days of onset of illness, however the exact number of days was not available for all patients. We apologize for not being able to include this information.

[Minor issues]

1. PLSDA cross-validation is necessary. Alternatively, authors may choose instead to use a method less prone to over-fitting, such as PCA or PCoA.

Reply:

We agree that that PLSDA is required to provide cross-validation for our data. We have consequently added PLSDA cross-validation results to Supplementary Fig. 17 in the revised manuscript.

PLS-DA cross validation details

Measure	1 comps	2 comps	3 comps	4 comps	5 comps
Accuracy	0.59303	0.70455	0.79576	0.79909	0.79909
R2	0.78665	0.91758	0.96615	0.98461	0.99485
Q2	0.63571	0.67457	0.67597	0.68082	0.68249

Supplementary Figure 17. PLS-DA cross-validation details for Fig. 1a

Plots obtained by the leave-one-out cross-validation (LOOCV) method applied to partial least squares discriminant analysis (PLS-DA) data. The PLS-DA cross-validation data showed cumulative values of $R^2 = 0.999$ and $Q^2 = 0.722$, indicating good clustering and good discrimination between the groups studied.

2. Page 5: “By contrast, levels of serum serotonin, which likely is released from the platelets, decreased (Supplementary Fig. 3a, b).” This statement is slightly misleading, since no significant difference in serotonin is observed between negative or asymptomatic patients vs severe patients.

Reply:

The Reviewer is correct in that serotonin levels are not statistically significantly reduced in the negative or asymptomatic group compared to the severe outcome group. These results (Supplementary Fig. 3a and 4b) and their description have been removed (Page 5).

Further downstream, levels of the deaminated metabolites—kynurenic, quinolinic, and nicotinic acid—were also elevated in the severe group. ~~By contrast, levels of serum serotonin, which likely is released from the platelets, decreased (Supplementary Fig. 3a, b).~~ In addition, threonine (Thr) levels were significantly decreased [...]

3. Supplementary figure 3b: significance stars appear to have been cut off.

Reply:

We thank the Reviewer for pointing this out, and have revised the figure accordingly.

4. Histidine and proline data in Supplementary Fig. 3 do not show strong differences between groups. I suggest that discussion of these either be tempered or removed.

Reply :

The Reviewer is correct; proline and histidine did not show a strong decrease ($p < 0.01$) in the severe group, as noted. Therefore, the wording has been softened as follows and proline data (Supplementary Fig. 3c) have been deleted (Page 6).

The levels of other amino acids ~~such as proline (Pro) and~~ including histidine (His) decreased in patients with poor prognosis, and those of urocanic acid, a product of His deamination, were significantly higher in the moderate and severe groups than in the other groups (Supplementary Fig. 3c).

On the other hand, the increase of urocanic acid is remarkable in moderate and severe cases. Histidine, which is the substrate of urocanic acid, exhibits a decreasing trend. Therefore, these results are worth presenting as an example of amino acid deamination. As this represents a key concern of this study, we have retained these results.

Supplementary Figure 3. Other characteristic changes in serum metabolites induced by COVID-19 early in disease correlate with future severity of pneumonia

Metabolomic analysis of serum from negative, asymptomatic, mild, moderate, and severe outcome patients (Cohort-1 samples) collected within 5 d of disease onset. Trp/Kynurenine ratio (a). Histidine and a deamino catabolite urocanic acid (b). AMP, a purine nucleotide (c) and hypoxanthine (d), a degradation product of purine nucleotides. Cysteine (e), Cystein sulfinat and cystein sulfinat/Cys ratio. Data are expressed as the mean value. Statistical significance was assessed using one-way ANOVA with Tukey's multiple comparisons test. * $p < 0.05$; ** $p < 0.01$; *** $p < 0.001$; **** $p < 0.0001$.

5. Figure 4a: it would be helpful to mark the metabolites listed onto the hierarchical clustering map, so that readers can assess the variability.

Reply :

We agree with the Reviewer's suggestion, and have highlighted the metabolites listed in the hierarchical clustering map as follows.

(Revised Figure)

6. Page 8 typo: “Although blood levels of the Leu did not change” should read “Although blood levels of Leucine did not change”

Reply :

The wording has been revised accordingly (Page 9).

Similar trends were observed for Gln (Fig. 4h, Supplementary Fig. 7g) and its deaminated product Glu (Fig. 4i, Supplementary Fig. 7h). Although blood levels of leucine did not change (Fig. 4J),

7. Page 8: “for example, levels of creatine and creatinine, which are thought to originate from muscle tissue 25, were elevated in the late stage of the disease (Fig. 4o)”. Data is only provided for creatine. Figure should be amended to provide creatinine data, or text should be modified.

Reply :

The Reviewer is correct. Since the creatinine data are not presented here, reference to them has been removed and the text changed to read as follows.

for example, levels of creatine, which is thought to be produced in the liver and transferred to muscle tissue²⁵, were elevated in the later stages of the disease (Figure 4o).

8. Page 22: “Fifty milliliters of microparticles, standards, and samples were added to the wells, followed by incubation for 2 h at room temperature.” Should this read “Fifty microliters”?

Reply :

The Reviewer is correct; this has been corrected in the revised manuscript.

9. Page 25: “Optical images of brain sections were obtained” should read: “Optical images of lung sections”

Reply :

We thank the Reviewer for pointing this out. We have added the corrected sentences to the Sample Preparation for Imaging Mass Spectrometry section.

Sample preparation for IMS

For IMS, lung tissues from heat-fixed SARS-CoV-2-infected or fresh-frozen influenza-infected mice were embedded in super cryoembedding medium (SCEM; SECTION LAB, Hiroshima, Japan) and then stored at -80 °C until analysis. The tissue-containing SCEM blocks were sectioned at a cryostat temperature of -16°C into 8-µm-thick sections using a CM 3050 cryostat (Leica, Wetzlar, Germany). These sections were transferred to indium tin oxide (ITO)-coated glass slides (Matsunami Glass Industries, Osaka, Japan) for further analyses. The mounted sections were manually coated with a matrix solution containing 9-aminoacridine (10 mg/mL dissolved in 80% ethanol) using an art brush (Procon Boy FWA Platinum, Mr. Hobby, Tokyo, Japan). The matrix solution was applied from a distance of approximately 15 cm, with approximately 1 mL sprayed on each slide. To maintain uniform conditions for analyte extraction and co-crystallization, matrix application was performed simultaneously on multiple slides, as previously described⁵. Optical images of lung sections were obtained by a scanner and subjected to MALDI-MS imaging.

10. Page 6: “the levels of glutamine (Gln), the amino acid with the highest blood concentration”. A reference would be helpful to support this statement.

Reply :

As suggested by the Reviewer, the following reference was cited in the relevant section to support the statements⁹.

11. Fig. 3a-f, please define a.u. in figure legend

Reply:

We thank the Reviewer for pointing this out. The Y-axis unit in the corresponding figure is Peak Area/IS (as in Fig. 2); the figure and legend have been modified accordingly.

Fig 3. Specific serum amino acid levels and their catabolite levels are prognostic predictors reflecting viral load during early COVID-19

Using samples from early COVID-19 onset (Cohort-1 samples), correlation analysis between the metabolite levels (peak area/IS) and PCR-Ct values was performed. Pearson correlation (r) and p-values are shown in the graphs (a–f and Table 2).

12. Table 3: please clarify the meaning of the bolding in Table 3.

Reply:

The Reviewer correctly points out that the bolded notation in Table 3 was not explained. In this table, metabolites that contribute to severe group discrimination by ROC analysis are listed in descending order of AUC. Those in bold are the amino acids and their catabolites that are the focus of this study and thus constitute the top candidates in the table, i.e., the discriminative markers of severe disease.

This has been clarified in the caption of Table 3 (Page 42).

Table 3. AUC and p-values for ROC curve analysis of metabolites potentially able to discriminate COVID-19 severe pneumonia patients from moderate, mild, and asymptomatic patients

Metabolites in bold represent amino acids and their catabolites that are the top candidates for discriminative markers of severe disease.

13. Figure 4: graphs showing the averages per-group would be helpful for Thr, Leu, Kyn, perhaps in Figure S4

Reply:

As per the Reviewer's suggestion, a graph has been added to S4 showing the averages for the Thr, Leu, and Kyn groups.

Reviewer #2 (Remarks to the Author):

Reply:

Thank you for reviewing our manuscript and for sharing Nature Communications' initiative to facilitate peer review training. We greatly appreciate the dedication and contribution of early career researchers in the review process.

We have carefully considered the feedback provided and have addressed each point. We invite you to review our responses to the comments and look forward to your further insights.

Reviewer #3 (Remarks to the Author):

The manuscript „Amino acid catabolite markers for early prognostication of pneumonia in patients with COVID-19“ by Maeda and co-workers examines an important question, namely early predictive markers and underlying biochemical pathomechanisms for a severe course of COVID-19.

By serum metabolome and cytokine analyses of COVID-19 patients (collected from multiple centers in Japan) changes of amino acid metabolism were identified as prognostic markers. Concentrations of several amino acids were correlated strongly with a high viral load, serum concentrations of the tryptophan catabolite kynurenine and adenyly succinate were suited best to differentiate between different courses of disease. Combined cytokine and metabolite analyses indicated a strong activation of the IFN γ -CXCL10/9 axis, which was then further examined by animal experiments.

In these animal experiments disease severity, lung metabolomic remodeling and cytokine production were investigated in mice infected with SARS-CoV2-MA10 and influenza virus. mRNA expression of Ifng, Cxcl10, Cxcl9, IDO-1, IDO-2, Bcat-1 and Ki67 was significantly higher in the severely affected Balb/c mice (both in mice infected with SARS-CoV2-MA10 and influenza virus) compared to the mild disease model- suggesting that the inflammatory response to SARS-CoV2-MA10 infection is similar to that in influenza infection.

Lung metabolomic analyses confirmed enhanced amino acid catabolism (along the kynurenine pathway and BCAA deamination) and de novo nucleotide synthesis pathway activity in the severely affected Balb/c mice compared to mildly affected B6 mice.

Abnormal proliferation of mucosal secretory cells and smooth muscle cells was found in both SARS-CoV2-MA10 and influenza mouse models, indicating similar remodeling of the lung in response to early respiratory virus infections.

The manuscript is of significance for the field and shows very interesting, novel data and should be published after minor modifications. The study design and the used methods are appropriate, the manuscript is written very well and the conclusions drawn are also supported by the data.

These are my questions/comments:

Reply:

We thank the Reviewer for their constructive comments.

We fully agree with the Reviewer's point that additional data are needed to understand the relationship between the early superficial symptoms seen in COVID-19 (e.g., general malaise) or the treatments required as symptoms worsen (e.g., airway intubation) and the blood metabolites that correlate with disease outcome. We also agree that additional clinical parameters that have already been reported (e.g., ferritin) should be provided. Therefore, we have provided as much additional clinical data as possible and added a correlation analysis between these data and metabolites, focusing on amino acids and their catabolites, as described in our response below.

On the other hand, our study presents clinical research data from a cohort of patients during the first, second, and initial phases of the third wave of COVID-19 in Japan from March to December 2020. While we believe that these are valuable data, it must be acknowledged that, during this period, hospitals faced a chaotic situation. Due to staff shortages and a lack of clinical resources, we were not able to conduct testing beyond routine clinical examinations. In addition, the availability of residual serum was limited, making additional measurements

difficult. Consequently, we are unable to provide all of the data requested, and would appreciate your understanding in this regard.

Nevertheless, we have added all supplemental data that are available at this time and hope that this revision will address the Reviewer's concerns.

- The number of patient samples with mild, moderate and severe course of disease of the different centres should be mentioned clearly in the main manuscript/Abstract. –

Reply:

We agree with the Reviewers' comment, and, in the revised manuscript have explicitly reported the number of patient samples classified as mild, moderate, and severe for each cohort, both in the body of the manuscript (Page 6, 9) and in the Abstract.

(Abstract) “Effective early-stage markers for predicting which patients are at risk of developing SARS-CoV-2 infection have not been fully investigated. Here, we performed comprehensive serum metabolome analysis of 83 patients from two cohorts to determine that the acceleration of amino acid catabolism within 5 d from disease onset correlated with future disease severity. Increased levels of de-aminated amino acid catabolites “

Page 6

“... To determine whether metabolite levels in the early stages of COVID-19 differed among the different outcome groups, we performed a metabolomic analysis of patient sera collected within 5 d of disease onset in Cohort-1, which comprised a total of 71 patients (17, 7, 17, 15 and 15 patients with negative, asymptomatic, mild, and moderate-to-severe outcomes, respectively).”

Page 9

“... To test whether the early metabolic changes that correlated with disease severity were specific to only early stages of disease, we analyzed sequential samples from Cohort-2, comprising a total of 12 patients (4 each with mild, moderate and severe outcomes, respectively).”

Actually, cohort-2 only includes 12 individuals, in which time course analyses were performed. Did time course analyses show similar patterns in men and women? (There appear to be gender-related differences regarding amino acid concentrations, which might be aggravated during severe disease)

Reply:

The Reviewers' comments regarding sex differences are both significant and enlightening. We have thoroughly reviewed the existing literature on sex differences in systemic amino acid metabolism in healthy humans and their potential implications for susceptibility to severe COVID-19 manifestations. Based on this, we have included a discussion that contextualizes our current data in relation to these differences.

First, it is worth noting that there are differences in blood amino acid profiles between healthy males and females. Specifically, concentrations of branched-chain amino acids (BCAAs) and their breakdown products have been found to be higher in males¹⁰. In addition, BCAA metabolism, along with the tyrosine and tryptophan pathways, was found to be significantly upregulated in males infected with COVID-19 compared to females¹¹. Although

it remains difficult to infer a causal relationship, the increased metabolism of BCAA and tryptophan may correlate with the observed poorer prognosis of male patients with COVID-19. Interestingly, our current investigation reveals that the catabolism of these amino acids is increased in association with a worse prognosis (Fig. 2). This phenomenon may be due to proliferation of lung parenchymal cells following an early phase of infection (Fig. 7). Consequently, it is possible that the higher metabolic response observed in male patients, specifically the increased breakdown of amino acids within the lung tissue, could contribute to the higher mortality rates from COVID-19 in males¹².

Within the cohort studied in this investigation, males predominantly made up the moderate and severe categories, which limits our ability to rigorously examine gender-related variations in blood amino acid levels. To illustrate, in Cohort 1, there was only 2 female patient in the severe category (out of a total of 15 patients) and 1 female in the moderate category (out of 15 patients). Cohort 2 had no female patients in its severe category, one female patient in the moderate group (out of 4 patients), and two in the mild category (out of a total of 4 patients). Such a distribution results in an inadequate sample size for meaningful statistical evaluation of gender differences. Attempts were also made to analyze the temporal variation of metabolite concentrations in relation to sex differences within cohort 2, consisting of 12 patients. However, once again, due to the limited sample size, definitive conclusions could not be drawn.

Nevertheless, we believe that this perspective warrants consideration in future research efforts, and we greatly appreciate the insightful feedback provided by the Reviewer. We expanded the Discussion section to emphasize the need for further research on gender differences in disease severity and the accompanying fluctuations in amino acid metabolism. In addition, we acknowledge the limitation posed by the underrepresentation of female participants in our cohort (Page 18).

There are several important reasons to further investigate the relationship between sex differences in human systemic amino acid metabolism and differences in the severity of clinical manifestations of COVID-19. First, it is apparent that amino acid profiles differ between men and women; in particular, the blood levels of BCAAs and subsequent catabolites are higher in men¹⁰. Furthermore, the existing literature emphasizes that BCAA metabolism is enhanced in men with COVID-19 compared with that in women; moreover, this pattern is similar for tyrosine and tryptophan catabolism¹¹. Our study also showed that increased catabolism of these amino acids correlates with poor prognosis (Fig. 2). The mechanism underlying this phenomenon is suggested to be the hyperproliferative response of lung parenchymal cells in the early stages of infection (Fig. 7). Although a direct causal relationship remains speculative, the increased metabolic reactivity observed in the lung parenchyma of male participants, particularly increased amino acid catabolism, may therefore contribute to the increased male mortality associated with COVID-19¹². A major limitation of our study is the predominance of men in the moderate and severe groups, which limited our ability to perform comprehensive sex-specific analyses. This distribution makes rigorous statistical investigation of sex differences difficult. Cohort-1 had one female patient in the severe category and one in the moderate category, while Cohort-2 had no females in the severe, one in the moderate, and two in the mild categories out of 4 patients for each group. This distribution makes a robust statistical evaluation of sex differences difficult. In particular, an analysis of Cohort-2 (comprising 12 patients) aimed at tracking the temporal progression of sex differences yielded inconclusive results due to sample size limitations. The

implementation of larger studies that account for sex differences is paramount to filling this knowledge gap.

- Please also state clearly, what the inclusion/exclusion criteria were: from the table it appears, that patients had maximally two/three co-morbidities.

Reply:

We agree with the Reviewer's comments, and have further clarified the inclusion and exclusion criteria for this study in the added Methods passage shown below (Page 20). The detailed characteristics of these cohorts are discussed in Methods-Participant section (Page 20).

“...Hospital and received negative RT-PCR test results, making them clinically unlikely to have COVID-19. The study included the early patient cohorts of waves 1, 2, and 3 of COVID-19 conducted in Japan from March to December 2020. For cohorts 1 and 2, patients were not excluded because of comorbidities, as the main objective was to include the greatest possible number of severe cases for which specimens were available early after disease onset. Cases in the mild and moderate groups were selected based on matching age and sex distribution with the severe group. No specific exclusions were made in any group during the study period, mainly because the number of samples was limited. Asymptomatic and mildly symptomatic patients were selected for inclusion by balancing the sex ratio. The higher proportion of males in the moderate and severe categories is due to the observed tendency for males to be more severely affected¹². This was particularly true for patients hospitalized during the study period.”

Actually, the number and also the extent of co-morbidities may of course also influence the course of disease- patients with more co-morbidities probably have more “background” inflammation/ higher stress (cortisol baseline) levels and secondary immunodeficiency might be a problem. You might e.g. calculate, whether there were differences regarding the investigated parameters in patients without/with co-morbidities- especially in patients with a more severe course of disease.

Reply:

The Reviewer correctly points out that it is necessary to verify whether the serum metabolites that vary with COVID-19 prognosis are influenced by the presence/type of comorbidity. In cohort 1, most patients with moderate or severe outcomes had some comorbidity. Therefore, we categorized comorbidities as follows and examined the relationship between their presence/absence and serum metabolite concentrations:

Infectious or non-infectious inflammatory and autoimmune diseases (n=3)

Atopic dermatitis, dermatomyositis, interstitial pneumonia, MPO-ANCA, asthma, ulcerative colitis, COPD, tuberculosis, Graves' disease.

Lifestyle-related diseases (n=6):

Diabetes mellitus, hypertension, dyslipidemia.

No complications or other diseases (n=4).

Patients with malignant tumors (n=2) were not examined due to the small sample size.

For each of the above complication categories, metabolite levels were compared to the "no complications, other" group. However, there were few statistically significant differences ($p > 0.01$) in metabolites correlated with COVID-19 severity, mainly due to the small sample size of each group.

The only major difference observed between patients with and without comorbidities was in the levels of serum PLP. As indicated in the main text, patients experiencing severe outcomes displayed a noteworthy decrease in serum PLP in comparison to patients with milder symptoms of COVID-19 (Fig. 2h). Furthermore, within the severe outcome group, PLP levels were dramatically lower in patients with inflammatory complications (Supplementary Fig. 5). This decrease may be caused by impaired liver function, resulting in decreased PLP production. Another possibility is increased PLP consumption by inflammatory tissues. Additionally, PLP itself inhibits the inflammasome¹³, so a significant decrease in PLP can create a vicious cycle that worsens excessive inflammation and worsens the severity of the disease.

These results are annotated in the text and details are added in Supplementary Information.

Page 7,

“We also examined whether serum metabolites correlated with COVID-19 prognosis were affected by the presence and type of comorbidities or by future treatment (oxygenation or airway intubation). While several metabolites were affected in this manner, this was not the case for any of the amino acids and their catabolites discussed here (Supplementary Fig.5).”

a Summary of comorbidities (Severe outcome)

Infectious or non-infectious inflammatory and autoimmune diseases (n=3)
ulcerative colitis, COPD, tuberculosis,
Lifestyle-related diseases (n=6):
Diabetes mellitus, Hypertension, Dyslipidemia.
No complications or other diseases (n=4)
Patients with Malignant tumors(n=2)

b Inflammatory disease

Lifestyle related disease

Supplementary Figure 5. The association between serum metabolite levels and comorbidities in the acute phase of COVID-19.

We tested whether serum metabolites, which are altered by COVID19 prognosis, are affected by the presence or type of comorbidities. In Cohort 1, most patients with moderate or severe outcomes had some comorbidity. Therefore, we categorized comorbidities (a) and examined the relationship between their presence or absence and serum metabolite concentrations (b): For each of the above complication categories, metabolite levels were compared to the "no complications, other" group. However, few statistically significant differences were found in metabolites correlated with COVID-19 severity ($p > 0.01$). The only significant difference observed between patients with and without

comorbidities was in serum PLP levels. As shown in the text, compared to patients with mild symptoms of COVID-19, patients with severe outcome showed a significant decrease in serum PLP (Fig. 2). Furthermore, among the severe outcome group, patients with inflammatory complications showed a dramatic decrease in PLP levels. This decrease may be due to decreased PLP production due to hepatic dysfunction or increased PLP consumption by inflammatory tissues. In addition, because PLP itself has an inhibitory effect on the inflammasome¹³, a marked decrease in PLP may lead to a vicious cycle that exacerbates excessive inflammation and promotes disease severity.

- According to the supplementary data presented also disturbances of hormones/neurotransmitters appear to be a major problem in Covid-19 infected patients with a more severe outcome: probably patients with a severe course of Covid-19 need higher cortisol levels to fight the infection/compensate, resulting in a reduced formation of other steroid hormones. Secondary adrenal insufficiency might in fact compromise the host response to infection significantly and might be the reason, why patients develop pneumonia/have a severe course of disease

Reply:

We appreciate the Reviewer's crucial point. As the Reviewer comments, it is important to note that fluctuations in bioactive molecules in the blood, i.e., increases and decreases in steroid hormones (Supplementary Fig. 2) and leakage of neurotransmitters into the blood (Fig. 4p, q), are observed early in the onset of COVID-19.

In particular, the trend noted by the Reviewer that "*patients with a severe course of Covid-19 need higher cortisol levels to fight the infection/compensate*" has been reported in previous reports¹⁴ and in the present study (Supplementary Fig. 2). Furthermore, in contrast to cortisol, testosterone was reported to be lower in the poor prognosis group¹⁵, which is also consistent with the trend found in the present study (Supplementary Fig. 2).

Although changes in steroid hormones in this study do not correlate as strongly with outcome as amino acids and their catabolites, the following considerations have been added to the Discussion (Page 7).

“It should also be noted that fluctuations in bioactive molecules in the blood, including increases or decreases in steroid hormones (Supplementary Fig. 2), were observed from the early onset of COVID-19. Patients with a severe course of COVID-19 have previously been reported to have increased cortisol¹⁴ and decreased testosterone¹⁵ levels from early infection onward, a trend consistent with that observed in this study.”

We thank the Reviewer again for their incisive comment.

- Did you also measure levels of steroid hormones in the mild/severe disease model? And did you also determine catecholamine levels and insulin levels in patients (Phenylalanine and tyrosine were correlated negatively with viral load, many patients developed prediabetes or diabetes and insulin is an antagonist of cortisol)?

Reply:

The Reviewer has a valid point. Unfortunately, we were not able to measure many of the parameters that the Reviewer highlights in this comment. A common reason for this is the confusion in the hospital during the sampling period, combined with a lack of staff and

clinical supplies. As a result, we were not able to perform extensive testing beyond routine clinical assessments. The specific reasons for each factor are detailed below. We also ask for your understanding that there is limited remaining serum, making it difficult to repeat all clinical tests.

Catecholamine levels:

Under regular medical practice (covered by Japan's National Health Insurance), there is no indication for measurement unless there is suspicion of conditions such as pheochromocytoma. Therefore, no measurement was performed in patients suspected of having COVID-19.

Insulin levels:

Similarly, there is no indication to measure in the absence of a suspicion of conditions such as insulinoma, and therefore no measurements were taken during visits of patients suspected of having COVID-19.

- The correlations between homocysteic acid, ADMA and viral load could indicate endothelial dysfunction. Did you also investigate other markers of endothelial dysfunction/microclot formation or oxidative/nitrosative stress? Oxidative stress is also induced by IFN-g, and depletion of antioxidants (like glutathione, vitamin C and vitamin E; but also of B vitamins- and vitamin B6 was low) can go along with impaired cell function and also impaired neurotransmitter formation. If you have data, please add or otherwise discuss.

Reply:

We thank the Reviewer for their productive suggestions. These comments are valid, and the correlation between several metabolites including homocysteic acid and ADMA and viral load shown in this paper may reflect endothelial dysfunction from early onset. However, laboratory data on blood markers that directly reflect endothelial dysfunction, microthrombus formation, and oxidative/nitrosative stress were not available, mainly because patients had no significant symptoms other than fever in the early stages of illness. We also apologize for not being able to generate additional results, as we were unable to secure the necessary sample volumes for additional measurements.

Furthermore, as the reviewer noted, a decrease in antioxidants may occur along with impairment of cellular function and neurotransmitter formation. The reduction of antioxidants (especially B vitamins - PLP) as markers of oxidative stress in our data is shown in Fig. 2h and is one of the main points of this study. On the other hand, no metabolomic data were available for other antioxidant metabolites, which may be due to the time between serum collection and freezing and storage conditions.

- Were there any differences regarding the investigated parameters between patients receiving oxygen treatment or intubation and those who did not?

Reply:

The Reviewer makes a pertinent observation. It is imperative to determine the presence of early blood metabolite markers associated with prognosis of respiratory function.

In response to this comment, we investigated whether the presence or absence of oxygen therapy or tracheal intubation in patients with poor prognosis affected the levels of candidate

amino acid catabolite prognostic markers associated with COVID-19 severity. The results show an absence of significant differences ($p>0.01$) for the amino acids and their catabolites highlighted in this paper, presumably due to the small sample size ($n=4$ and 10 , for non-intubation and intubation, respectively). As a representative example, Trp and Thr and their respective catabolites showed no significant differences with or without oxygenation and airway intubation.

On the other hand, serum concentrations of 2,3-diphosphoglycerate (2,3-DPG), which is highly concentrated in erythrocytes¹⁶, were significantly lower in patients who later underwent airway intubation. This finding may be attributed to altered erythrocyte metabolism in patients who will require intubation in the future, resulting in lower serum levels of 2,3-DPG.

Again, the small sample size does not allow us to draw strong conclusions, but we have noted this in the revised Discussion as a finding that requires further study.

These results are annotated in the text (Page 9), and details have been added as Supplementary Fig. 6.

a Tracheal intubation (Severe outcome group)

b Oxygen administration (Moderate outcome group)

Supplementary Figure 6. The association of serum metabolite levels of COVID-19 in the acute phase with treatment during severe pneumonia.

We examined whether the presence or absence of tracheal intubation (a) or (b) in poor prognosis patients impacted the concentration of potential amino acid catabolite indicators for COVID-19 severity. The findings indicated no statistically significant variances ($p > 0.01$) among the amino acids and their catabolites discussed in this manuscript, possibly due to the limited sample size ($n = 4$ and 10 for non-intubated and intubated patients, respectively). As an example, there were no significant differences in Trp and Thr and their respective catabolites between the oxygenation and airway intubation groups. However, patients who underwent airway intubation had significantly lower serum concentrations of 2,3-diphosphoglyceric acid (2,3-DPG), which is highly concentrated in red blood cells¹⁶. The lower serum 2,3-DPG concentrations¹⁶ in patients requiring future intubation may be the result of altered metabolism in their red blood cells.

And did you record other symptoms patients had during acute Covid-19 (e.g. neurological symptoms, chest pain, dyspnea, strong fatigue, sleep disturbances, depression/anxiety, pain,...)? If yes, were there any differences regarding amino acids in patients with/without symptoms?

(Enhanced tryptophan catabolism can impair serotonin and melatonin formation and serotonin levels were lower in patients with more severe disease, this might explain many symptoms that hospitalized patients, but also Post Covid patients have, e.g. sleep disturbance, depressed mood, anxiety, fatigue).

Reply:

We strongly agree with the Reviewer' comment. The association of serum metabolite levels with surface symptoms during acute COVID-19 is an important consideration. Consequently, we reviewed the clinical information for Cohort 1 and added the presence or absence of the following symptoms. (see below and Supplementary Fig. 4a);

1. Neurological/psychiatric symptoms
2. Chest pain
3. Dyspnea
4. General malaise

A correlation analysis (chi-squared test) between the frequency of occurrence of these symptoms and the COVID-19 outcome showed that only general malaise was significantly correlated, suggesting that the prognosis is poor with the onset of general malaise. A high viral load is suggested in the poor prognosis cases in Cohort 1 of this study (Fig. 3), which may be associated with the induction of generalized malaise.

	asymptomatic	mild	moderate	severe	Chi-squared sum	p-value
General malaise (%)	0	5.9	41.2	47.1	11.5	0.009
Dyspnea (%)	0	23.5	29.4	23.5	2.5	0.471
Chest pain (%)	0	5.9	29.4	11.8	5.6	0.135
Neurological/psychiatric symptoms (%)	14.3	0	5.9	17.6	3.8	0.283

On the other hand, the presence or absence of general malaise was not significantly connected to the levels of amino acids and their catabolites that correlate with COVID-19 prognosis.

Trp and Kyn are shown below as representative examples. Similar results were confirmed for the presence or absence of dyspnea. Thus, the presence or absence of superficial symptoms does not appear to be reflected in differences in blood metabolite levels early in the course of the disease.

(general malaise)

These results are annotated in the text and details have been added as Supplementary Fig. 4.

	asymptomatic	mild	moderate	severe	Chi-squared sum	p-value
General malaise (%)	0	5.9	41.2	47.1	11.5	0.009
Dyspnea(%)	0	23.5	29.4	23.5	2.5	0.471
Chest pain(%)	0	5.9	29.4	11.8	5.6	0.135
Neurological/ psychiatric symptoms (%)	14.3	0	5.9	17.6	3.8	0.283

General malaise

Supplementary Figure 4. The association between serum metabolite levels and superficial symptoms in the acute phase of COVID-19.

Patients in Cohort-1 were assessed for the presence of the following superficial symptoms;

1. Neurological/psychiatric symptoms, 2. Chest pain, 3. Dyspnea, 4. General malaise.

Correlation analysis (chi-squared test) between the frequency of occurrence of these symptoms and COVID-19 outcomes showed that only general malaise was significantly correlated (a), suggesting that the occurrence of general malaise was associated with poorer prognosis. The higher viral load in the poor prognosis cases in Cohort-1 of this study (Fig. 3) suggests a possible association with the induction of general malaise. On the other hand, the presence of general malaise was not significantly related to the levels of amino acids and their catabolites which correlated with the prognosis of

COVID-19 shown in this study (b). Similar results were confirmed for the presence of dyspnea. Thus, the presence or absence of superficial symptoms does not appear to be reflected in differences in blood metabolite levels early in the course of the disease.

- Please also provide data of hemoglobin, iron parameters, serum interleukin-6, ferritin and ideally neopterin concentrations of patients (if available) and in case discuss them. (Earlier studies suggested that these parameters were also predictive for a worse outcome of patients: e.g. <https://doi.org/10.3390/metabo11100653>; <https://doi.org/10.1021/acs.jproteome.1c00052>; <https://doi.org/10.1186/s12879-020-05671-7>; <https://doi.org/10.1093/ofid/ofaa521>; <https://doi.org/10.1016/j.jphs.2022.06.005>;

Reply:

The Reviewer provides an important comment; these data should be considered for previously reported adverse outcome factors, including iron-related parameters. Following this suggestion, additional data regarding hemoglobin and ferritin were obtained for Cohort-1 (Supplementary Fig. 16).

There was no difference in hemoglobin between the patient groups at early onset, at the same time as the metabolomic measurements. In contrast, there was a predominant increase in ferritin in the severe group compared to the mild group. In this study, the degree of correlation between increased ferritin concentration and worse prognosis was weaker than in the metabolome.

These findings are consistent with those of previous reports. In particular, ferritin elevation tends to be attenuated in the later stages of COVID-19, similar to the dynamics of the metabolite in the present study. These observations were added to Supplementary Fig. 16.

Supplementary Figure 16. Hemoglobin and ferritin levels for Cohort-1

At early onset, contemporaneous with the metabolome measurements, there were no differences in hemoglobin between patient groups (a). In contrast, there was a predominant increase in ferritin in the severe disease group compared to the mild disease group (b).

However, it was difficult to generate results for the other parameters indicated by the Reviewer, because data were not obtained at the time of admission and additional analysis was not straightforward.

IL6: at the time of admission of COVID-19 patients, blood IL6 testing was not covered by standard medical care (covered by the National Health Insurance of Japan), and due to staff shortage at the hospital at the time of the second wave, no such data were collected.

Other iron parameters: these are not tested under standard medical care except when anemia is suspected and a differential diagnosis is made.

Neopterin: this also is not tested in routine medical care. Further, it is not quantifiable by LCMS (a HPLC fluorescence detector is usually employed), and, consequently, no data were obtained in this study.

-I would also suggest to discuss very recent papers investigating metabolomic approaches in patients with acute Covid-19 and the role of IDO in orchestrating immune response.

Reply:

In accordance with the Reviewer's comments, we have referenced and discussed the latest metabolomics approaches in acute COVID-19 patients, including a recent article investigating the role of IDO in the regulation of immune responses as follows;

(Page 15)

“The fact that both CXCL10 production and amino acid catabolism were reduced in the late phase of the disease (Fig. 4), when viral load was reduced¹⁷, also supports this hypothesis. Very recently, an increased KYN/TRY ratio due to IDO has also been reported in patients with severe COVID¹⁸. Since IDO and its catabolites kynurenines suppress T-cell responses and promote Treg proliferation^{19, 20} this may be a strategy of the virus to induce immune tolerance.”

Reviewer #4 (Remarks to the Author):

The experimental studies in mice are nicely done and appropriate controls are included. The conclusions relating to remodeling of lung tissue and hyperproliferation of certain cells in lungs are very convincing. While the comparison between high- and low-dose infection is legitimate it is unclear how any of this impacts the virus dissemination in lung tissue. The reader would benefit from being able to look at the virus load data in form of "infectious virus tissue titration" and not only immunohistology. At this stage it is unclear if there is potentially any antiviral effect not predicted in some of the groups. In other words, is there a difference in virus growth that contributes to pathological differences in these lungs? Otherwise, this is a very solid study that increases our understanding of corona virus pathology.

Reply:

We appreciate the Reviewer's insights into viral proliferation and pathological changes in the lung.

We are of the opinion that in the experimental infection with SARS-COV2-MA10 in this study, the viral dose positively correlated with the actual viral spread in the lungs, resulting in morphological and metabolic changes in the lung tissue. However, as the Reviewer points out, data need to be provided to show that the viral dose positively correlates with actual viral replication in the lung.

Consequently, the level of viral proliferation was assessed by single-cell RNAseq in mouse lung models with high and low levels of infection. Specifically, we compared the amount of virus infecting cells obtained from infected lung tissue (Gate on CD45- population) with the amount of virus infecting CD45 cells. This method detected viruses in cells other than immune cells, but not viruses that had not infected cells.

The results show that all genes measured were significantly elevated at the higher doses, and indicate that viruses proliferate in lung parenchymal cells in a dose-dependent manner. We hope that these data address the Reviewer's concerns.

Genes Name	Mild Average	Mild Log2 Fold Change	Severe Average	Severe Log2 Fold Change
N	2.99	-3.56	35.33	3.56
M	0.32	-3.50	3.57	3.50
E	0.03	-3.74	0.35	3.74
ORF1ab	0.50	-4.00	7.92	4.00
ORF10	0.24	-3.64	2.99	3.64
ORF7a	0.22	-3.78	3.05	3.78

Detection of virus genes in lung tissue of CD45-population infected mice

Balb/c mice were nasally infected with Mild (3E+02 TCID50/mice) or Severe (3E+04 TCID50/mice) doses of SARS-CoV2 (MA10 strain). One day later, after perfusing mouse lungs to create single-cell suspensions, CD45- cells were isolated by cell sorting (CytoFLEX SRT, BECKMAN COULTER). The feature expression profiles were extracted from the output matrix file generated from 10x Genomix Cellranger (doi:10.1038/ncomms14049). The data were analyzed using Loupe Browser (10x Genomics). The above Table shows an excerpt of the SARS-CoV2 genes.

References

1. Svensson, M. et al. Heat Stabilization of the Tissue Proteome: A New Technology for Improved Proteomics. *Journal of Proteome Research* **8**, 974-981 (2009).
2. Cazares, L. H. et al. Heat fixation inactivates viral and bacterial pathogens and is compatible with downstream MALDI mass spectrometry tissue imaging. *BMC Microbiology* **15**, (2015).
3. Saigusa, D. et al. Simultaneous Quantification of Sphingolipids in Small Quantities of Liver by LC-MS/MS. *Mass Spectrometry* **3**, S0046-S0046 (2014).
4. Sugiura, Y., Honda K., Kajimura M., Suematsu M. Visualization and quantification of cerebral metabolic fluxes of glucose in awake mice. *Proteomics* **14**, 829-838 (2014).
5. Sugiura, Y., Setou M., Horigome D. Methods of Matrix Application. In: *Imaging Mass Spectrometry*. Springer Japan (2010).
6. Sugiura, Y. et al. Visualization of in vivo metabolic flows reveals accelerated utilization of glucose and lactate in penumbra of ischemic heart. *Sci Rep* **6**, 32361 (2016).
7. Zhang, B. et al. B cell-derived GABA elicits IL-10. *Nature* **599**, 471-476 (2021).
8. Chong, J. et al. MetaboAnalyst 4.0: towards more transparent and integrative metabolomics analysis. *Nucleic Acids Res* **46**, W486-W494 (2018).
9. Cruzat, V., Macedo Rogero M., Noel Keane K., Curi R., Newsholme P. Glutamine: Metabolism and Immune Function, Supplementation and Clinical Translation. *Nutrients* **10**, 1564 (2018).
10. Krumsiek, J. et al. Gender-specific pathway differences in the human serum metabolome. *Metabolomics* **11**, 1815-1833 (2015).
11. Escarcega, R. D. et al. Sex differences in global metabolomic profiles of COVID-19 patients. *Cell Death & Disease* **13**, (2022).
12. Chanana, N. et al. Sex-derived attributes contributing to SARS-CoV-2 mortality. *Am J Physiol Endocrinol Metab* **319**, E562-E567 (2020).
13. Zhang, P. et al. Vitamin B6 Prevents IL-1 β Protein Production by Inhibiting NLRP3 Inflammasome Activation. *Journal of Biological Chemistry* **291**, 24517-24527 (2016).
14. Amiri-Dashatan, N., Koushki M., Parsamanesh N., Chiti H. Serum cortisol concentration and COVID-19 severity: a systematic review and meta-analysis. *J Investig Med* **70**, 766-772 (2022).
15. Camici, M. et al. Role of testosterone in SARS-CoV-2 infection: A key pathogenic factor and a biomarker for severe pneumonia. *Int J Infect Dis* **108**, 244-251 (2021).
16. Macdonald, R. Red cell 2,3-diphosphoglycerate and oxygen affinity. *Anaesthesia* **32**, 544-553 (1977).
17. He, X. et al. Temporal dynamics in viral shedding and transmissibility of COVID-19. *Nature Medicine* **26**, 672-675 (2020).
18. Al-Hakeim, H. K., Khairi Abed A., Rouf Moustafa S., Almulla A. F., Maes M. Tryptophan catabolites, inflammation, and insulin resistance as determinants of chronic fatigue syndrome and affective symptoms in long COVID. *Front Mol Neurosci* **16**, 1194769 (2023).
19. Benavente, F. M. et al. Contribution of IDO to human respiratory syncytial virus infection. *J Leukoc Biol* **106**, 933-942 (2019).
20. Mehraj, V., Routy J. P. Tryptophan Catabolism in Chronic Viral Infections: Handling Uninvited Guests. *Int J Tryptophan Res* **8**, 41-48 (2015).

Reviewers' Comments:

Reviewer #1:

Remarks to the Author:

Overall all our comments have been satisfactorily addressed. Two of the edited sections may contain typos:

1. "through a gradient elution with mobile phase A (0.1% formate) and mobile phase B (0.1% acetonitrile)." Can the authors verify that they did indeed use 0.1% acetonitrile for mobile phase B? Mobile phase B is usually a much higher acetonitrile percentage, from 80-100% usually, plus 0.1% formate. Was mobile phase A water plus 0.1% formate? Current phrasing is a bit confusing.
2. "Analyses were performed using Metaboanalyst (v4.0), a web-based multivariate analysis tool⁵⁹ For normalization". Period is missing "after multivariate analysis tool⁵⁹".

Reviewer #2:

Remarks to the Author:

Reviewer #3:

Remarks to the Author:

The quality of the manuscript has improved a lot by the revision, the paper can be published in the current form.

All my questions have been answered (unfortunately not all requested laboratory data of interest were available- but due to reasonable causes) and relevant data been added or discussed.

The only thing that I would change is replace "5 d" by 5 days in the whole manuscript.

Reviewer #4:

Remarks to the Author:

Authors have addressed all raised issues.

We would like to thank the reviewers for their comprehensive and constructive feedback. The recommendations and comments received on two occasions have improved the quality of the manuscript and all concerns raised during the review process have been addressed.

Reviewer #1 (Remarks to the Author):Overall all our comments have been satisfactorily addressed. Two of the edited sections may contain typos:

1. "through a gradient elution with mobile phase A (0.1% formate) and mobile phase B (0.1% acetonitrile)." Can the authors verify that they did indeed use 0.1% acetonitrile for mobile phase B? Mobile phase B is usually a much higher acetonitrile percentage, from 80-100% usually, plus 0.1% formate. Was mobile phase A water plus 0.1% formate? Current phrasing is a bit confusing.
2. "Analyses were performed using Metaboanalyst (v4.0), a web-based multivariate analysis tool⁵⁹ For normalization". Period is missing "after multivariate analysis tool⁵⁹".

Reply.

The reviewer correctly pointed out that the correct description is "acetonitrile containing 0.1% formic acid," and this has been corrected.

We have also added the period that had been omitted after "multivariate analysis tool⁵⁹."

Reviewer #2 (Remarks to the Author):I co-reviewed this manuscript with one of the reviewers who provided the listed reports. This is part of the Nature Communications initiative to facilitate training in peer review and to provide appropriate recognition for Early Career Researchers who co-review manuscripts.

Reviewer #3 (Remarks to the Author):The quality of the manuscript has improved a lot by the revision, the paper can be published in the current form. All my questions have been answered (unfortunately not all requested laboratory data of interest were available- but due to reasonable causes) and relevant data been added or discussed. The only thing that I would change is replace "5 d" by 5 days in the whole manuscript.

Reply.

As pointed out by the reviewer, 5d in the text has been replaced to 5days.

Reviewer #4 (Remarks to the Author):
Authors have addressed all raised issues.